# Provably Label-Efficient Conformal Prediction

**Andrew Ilyas** [1]   **Joonhyuk Ko** [2]   **Jingwu Tang** [1]   **Zhiwei Steven Wu** [1]   **Jiahao Zhang** [1]

## Abstract

Conformal prediction converts any black-box predictor into one with finite-sample, distribution-free coverage guarantees, outputting prediction sets $T(x)$ that contain the true label with probability at least $1 - \alpha$. To construct these prediction sets, conformal prediction relies on a randomly sampled "calibration set" of labeled examples. In many applications, however, this labeled calibration set is costly to collect, creating a tradeoff between upfront labeling cost and downstream utility of the conformal predictor. In this work, we study *conformal prediction with costly label queries*, where unlabeled examples arrive i.i.d. and labels can be queried one at a time. After $m$ queries, we form a conformal predictor; the upfront cost of this predictor is the calibration set size $m$, and its efficiency is the expected prediction set size $\mathbb{E}|T_m(X)|$. We design an online stopping rule $\hat{m}$ that automatically balances the upfront cost against conformal efficiency *while preserving the original conformal guarantee*. Theoretically, we show that under mild regularity assumptions, the expected total cost of our stopping rule matches the best fixed calibration size in hindsight. Experimentally, we find that our stopping rule reduces cost compared to standard choices of $m$ from the literature by $40.6\% \pm 2.3\%$. Finally, we demonstrate a reduction from the probably approximately correct labeling problem of Candès et al. (2025) to CP, under which our stopping rule minimizes the total labeling cost.

---

[1]School of Computer Science, Carnegie Mellon University [2]Department of Computer Science, Princeton University. Correspondence to: Andrew Ilyas <andrewi@andrew.cmu.edu>, Joonhyuk Ko <jk9737@princeton.edu>, Jingwu Tang <jingwutang@cmu.edu>, Zhiwei Steven Wu <zstevenwu@cmu.edu>, Jiahao Zhang <jiahaozhang@cmu.edu>.

*Proceedings of the 43$^{rd}$ International Conference on Machine Learning*, Seoul, South Korea. PMLR 306, 2026. Copyright 2026 by the author(s).

## 1. Introduction

Conformal prediction (CP) has emerged as a rigorous framework for uncertainty quantification, providing finite-sample, distribution-free correctness guarantees about black-box machine learning models. Consider a distribution $\mathcal{D}$ over the domain $\mathcal{X} \times \mathcal{Y}$, where $\mathcal{X}$ is the covariate space and $\mathcal{Y}$ is the label space. Using a set of calibration samples $(X_1, Y_1), ..., (X_m, Y_m)$ drawn i.i.d. from $\mathcal{D}$, the objective of conformal prediction is to create a prediction set $C(x)$, for each input $x$, that is likely to include the true label $y$. This is formalized through specific coverage guarantees on the prediction sets. For example, the simplest and most commonly-used guarantee is marginal coverage: The prediction sets $T(x) \subset \mathcal{Y}$ achieve marginal coverage if, for a test sample $(X_{m+1}, Y_{m+1})$, we have $\Pr(Y_{m+1} \in T(X_{m+1})) \geq 1 - \alpha$. Here, $\alpha$ is the *miscoverage rate*, and the probability is taken over the randomness in the calibration and test points.

While the validity of the coverage guarantee holds regardless of the calibration set size, the efficiency of the resulting prediction sets—as measured, e.g., by their expected size—improves as the calibration set grows. This creates a fundamental tension in applications where high-quality labels are scarce or expensive. For instance, in medical diagnostics, obtaining a ground truth label may require invasive procedures or a consensus of expert specialists. Similarly, in social science or policy research, high-quality annotation often necessitates extensive human effort. In these "costly-label" regimes, the standard practice of using a fixed, arbitrarily chosen calibration set is suboptimal: a set that is too small yields uninformative (large) prediction sets, while a set that is too large incurs unnecessary labeling costs.

In this work, we study conformal prediction with costly label queries, a setting where we seek to minimize the total cost of a conformal prediction pipeline: the sum of the labeling cost incurred to build the calibration set and the inefficiency cost of the resulting conformal predictor. We model this as a sequential problem where unlabeled examples arrive i.i.d., and we must decide when to stop querying labels to form our conformal predictor. This problem is distinct from recent work on length optimization in conformal prediction, which focuses on optimizing the set construction for a fixed calibration set (Kiyani et al., 2024), effectively ignoring the cost of acquiring that set. It also differs from active,

anytime-valid risk control, which focuses on maintaining valid risk guarantees continuously over a stream of data with an active labeling policy. In contrast, our goal is to identify the optimal calibration size $\hat{m}$ that strikes the best trade-off between labeling effort and downstream predictive efficiency, effectively automating the "budgeting" decision for conformal calibration.

**Contributions and roadmap.** Our work aims to initiate the study of end-to-end cost-sensitive conformal prediction.

1. In Section 3, we formalize the problem of conformal prediction with costly label queries;
2. In Section 4, we propose a cost-sensitive stopping rule that trades off the marginal cost of acquiring a label against the expected reduction in prediction set size;
3. In Section 5, we study this stopping rule theoretically and show that it achieves *no regret*; that is, the total expected cost of our stopping rule matches the best fixed stopping rule in hindsight up to lower-order terms (see Definition 3.3);
4. In Section 6, we demonstrate a reduction from cost-sensitive PAC Labeling, as introduced by Candès et al. (2025), to cost-sensitive conformal prediction, and that under this reduction, our stopping rule exactly minimizes the total expected labeling cost;
5. In Section 7, we empirically validate our theory on standard conformal prediction and PAC Labeling benchmarks; we find that using our stopping rule consistently saves cost over the "default" calibration set sizes, and indeed attains similar performance to the best fixed calibration set in hindsight.

## 2. Related Work

Our work builds on several parallel lines of work across statistics and machine learning—we highlight the closest connections here.

**Length efficiency in conformal prediction.** There is a growing body of work on designing improved nonconformity scores to enhance length efficiency in conformal prediction (Chernozhukov et al., 2021; Deutschmann et al., 2023; Feldman et al., 2021; Lei et al., 2018; Xie et al., 2024; Yang & Kuchibhotla, 2025; Papadopoulos et al., 2011; Romano et al., 2020). Kiyani et al. (2024) further studied the problem of length optimization under conditional validity constraints. Xu et al. (2024) considered a similar problem termed as risk controlling prediction set. However, these works do not account for the tradeoff between predictive accuracy and the cost of acquiring a calibration dataset. In contrast, we explicitly model this tradeoff by minimizing a joint objective that combines the labeling cost of constructing the calibration set with the inefficiency of the resulting conformal predictor.

**Adaptive dataset labeling.** Our framework is applicable to the adaptive labeling problem studied by Candès et al. (2025), which lies within the broader literature on efficient dataset labeling from potentially noisy or weak supervision. Much of the existing work in this area either relies on strong parametric or distributional assumptions (Qiu et al., 2020; Ratner et al., 2016; Northcutt et al., 2021) or lacks formal accuracy guarantees (Bernhardt et al., 2022; Iscen et al., 2019; Li et al., 2023; Xie et al., 2020; Zhu & Ghahramani, 2002). While Candès et al. (2025) provides provable accuracy guarantees, it does not offer formal guarantees on labeling cost. In contrast, our framework explicitly accounts for labeling costs and provides a principled trade-off between labeling efficiency and predictive performance, with provable guarantees on both labeling cost and predictive accuracy.

## 3. Problem Formulation

We consider the multi-class prediction setting where the label space satisfies $|\mathcal{Y}| = K$. Data pairs $(X, Y)$ are drawn i.i.d. from an unknown distribution $\mathcal{D}$ over $\mathcal{X} \times \mathcal{Y}$. Our goal is to construct prediction sets $T(x) \subseteq \mathcal{Y}$ that are both *valid*—in the sense of achieving a prescribed miscoverage level—and *efficient*, as measured by their expected size, while accounting for the cost of querying true labels. We begin by reviewing split conformal prediction, which provides distribution-free coverage guarantees. We then formulate our main problem: choosing how many labels to query in order to minimize a total cost, subject to a high-probability coverage constraint.

### 3.1. Split Conformal Prediction

Split conformal prediction is a standard framework for constructing prediction sets with coverage guarantees (Vovk et al., 2005; Romano et al., 2019). The method relies on a nonconformity score function $s : \mathcal{X} \times \mathcal{Y} \to \mathbb{R}$, which quantifies how incompatible a candidate label $y$ is with an input $x$. For instance, given an estimate $\hat{p}(y \mid x)$ of the conditional distribution $\Pr(Y = y \mid X = x)$, a common choice is the log-loss score $s(x, y) = -\log \hat{p}(y \mid x)$.

Given a calibration dataset $D_m = \{(X_i, Y_i)\}_{i=1}^m$, split conformal prediction computes a threshold $\hat{q}_m$ and outputs the prediction set $T_{D_m}(x) = \{y \in \mathcal{Y} : s(x, y) \leq \hat{q}_m\}$. Intuitively, $\hat{q}_m$ is chosen to approximate a $(1 - \alpha)$-quantile of the scores, so that labels whose scores are sufficiently small are included in the prediction set. Algorithm 1 gives a high-probability variant based on concentration inequalities. The following theorem states the resulting coverage guarantee, also called *training-conditional coverage*[1].

---

[1]A more classical formulation of split conformal prediction selects $\hat{q}_m$ as the $\lceil (m+1)(1-\alpha) \rceil$-th order statistic of the calibration scores, equivalently the empirical $\lceil (m+1)(1-\alpha) \rceil / m$-quantile,

**Algorithm 1** Split Conformal Prediction

**Input:** calibration dataset $D_m = \{(X_i, Y_i)\}_{i=1}^m$, nonconformity score function $s(\cdot)$, target miscoverage rate $\alpha$, failure probability $\delta$.

1: Set the confidence bonus: $r_m = \sqrt{\frac{\log(2/\delta)}{2m}}$.
2: Let $\hat{q}_m$ be the smallest value such that

$$\frac{1}{m}\sum_{i=1}^m \mathbf{1}[s(X_i, Y_i) \le \hat{q}_m] \ge (1-\alpha) + r_m. \quad (1)$$

**Output:** Prediction set $T_{D_m}(x) = \{\hat{y} \in \mathcal{Y} : s(x, \hat{y}) \le \hat{q}_m\}$.

---

**Theorem 3.1** (High-Probability Split Conformal Guarantee (Vovk, 2012))**.** *Fix any distribution $\mathcal{D}$ over $\mathcal{X} \times \mathcal{Y}$, any target miscoverage $\alpha \in [0, 1]$, and any nonconformity score function $s$. Let $D_m \sim \mathcal{D}^m$. Then, with probability at least $1 - \delta$ over the draw of $D_m$,*

$$\Pr_{(X,Y)\sim\mathcal{D}}\big(Y \in T_{D_m}(X)\big) \ge 1 - \alpha.$$

### 3.2. Cost Minimization with Coverage Constraints

We now formulate the problem of *conformal prediction with costly label queries*. Suppose we observe a stream of $n$ unlabeled feature vectors. To construct a conformal predictor, we must decide how many labels to query. Querying one label incurs a unit cost, and querying $m$ labels yields a calibration set $D_m$.

Using $D_m$, we construct the conformal predictor $T_{D_m}$ via Algorithm 1. We define the *total expected cost* as

$$C(m) = m + (n-m) \cdot \mathbb{E}_{(X,Y)\sim\mathcal{D},\, D_m\sim\mathcal{D}^m}\left[\frac{|T_{D_m}(X)|}{K}\right]. \quad (2)$$

The first term accounts for the cost of querying $m$ true labels. The second term measures the inefficiency of the resulting predictor on the remaining $n - m$ points, quantified by the expected size of the prediction set.

The normalization by $1/K$ places both terms on a common scale. Indeed, a maximally uninformative prediction corresponds to the full label set $\mathcal{Y}$, which has size $K$ and hence incurs a normalized cost of 1, which is the same as querying an expert label. Conversely, querying a true label can be interpreted as paying one unit of cost to replace a maximally inefficient prediction set by a perfectly informative

---

which guarantees marginal coverage *in expectation* over the randomness of $D_m$: $\mathbb{E}_{D_m}\big[\Pr_{(X,Y)\sim\mathcal{D}}\big(Y \in T_{D_m}(X)\big)\big] \ge 1-\alpha$. In contrast, throughout this work we adopt the high-probability variant above, which ensures coverage holds for a fixed realization of the calibration dataset, a property that is essential for cost-sensitive and adaptive label acquisition.

---

singleton. Under this interpretation, equation 2 represents the total cost of resolving uncertainty across the dataset. For continuous label spaces, an analogous formulation replaces normalized set cardinality with normalized set measure; see Section B for a discussion.

We now formalize the decision process.

**Definition 3.2** (Stopping Rule)**.** A general stopping rule is defined as a sequence of functions $\phi = (\phi_m)_{m=1}^n$, where $\phi_m : (\mathcal{X} \times \mathcal{Y})^m \to \{0, 1\}$ indicates whether to stop label acquisition:

$$\phi_m(D_m) = \begin{cases} 1 & \text{if we stop at or before step } m, \\ 0 & \text{otherwise.} \end{cases} \quad (3)$$

The induced stopping time (calibration size) is then given by:

$$\hat{m} = \inf\{m \ge 1 : \phi_m(D_m) = 1\}. \quad (4)$$

Now we define no-regret stopping rule.

**Definition 3.3** (Regret)**.** Let $m^\star = \arg\min C(m)$ be the minimizer of the total expected cost. Define the regret of a stopping rule $\phi$ as

$$R(\phi) := \mathbb{E}[C(\hat{m})] - C(m^\star),$$

where the expectation is taken over the randomness of $\hat{m}$.

And we say a stopping rule is *no-regret* if

$$\lim_{n\to\infty} \frac{R(\phi)}{C(m^\star)} = 0.$$

Our goal is to design a stopping rule that randomly selects a calibration size $\hat{m} \in \{1, \ldots, n\}$ and constructs a conformal predictor $T_{D_{\hat{m}}}$ such that:

1. **Validity:** The predictor achieves miscoverage at most $\alpha$ with high probability $1 - \delta$;

2. **No regret:** The total expected cost of our stopping rule matches that of the best fixed stopping rule in hindsight up to lower-order terms.

To this end, we will introduce a stopping rule in the next section that adaptively select $\hat{m}$ and the corresponding threshold $\hat{q}_{\hat{m}}$ to meet these guarantees. Notably, our analysis extends beyond equation 2 to any objective formulated as a linear combination of labeling cost and prediction-set inefficiency.

## 4. No-regret Stopping Rule

In this section, we propose a simple stopping rule and show that, under a mild monotonicity assumption, it is no-regret.

**Algorithm 2** Anytime-Valid Split Conformal Prediction with Stopping Rule

---

**Input:** Unlabeled stream $(X_1, \ldots, X_n)$, score $s(\cdot, \cdot)$, target miscoverage $\alpha \in (0,1)$, confidence budget $\delta \in (0,1)$, stopping rule $\phi$.

1: Initialize $D_0 \leftarrow \emptyset$.
2: **for** $m = 1$ to $n$ **do**
3:   Query $Y_m$ and set $D_m \leftarrow D_{m-1} \cup \{(X_m, Y_m)\}$.
4:   Set $p_m \leftarrow (1 - \alpha) + r_m$.
5:   **if** $p_m \geq 1$ **then**
6:     Set $\hat{q}_m \leftarrow +\infty$.
7:   **else**
8:     Let $\hat{q}_m$ be the smallest value such that

$$\frac{1}{m} \sum_{i=1}^{m} \mathbf{1}\big[s(X_i, Y_i) \leq \hat{q}_m\big] \geq p_m.$$

9:   **end if**
10:   Define the time-$m$ predictor $T_m(x) := \{\hat{y} \in \mathcal{Y} : s(x, \hat{y}) \leq \hat{q}_m\}$.
11:   **if** $\phi_m(D_m) = 1$ **then**
12:     **break**
13:   **end if**
14: **end for**
15: Let $\hat{m} := \inf\{m \geq 1 : \phi_m(D_m) = 1\} \wedge n$ be the induced stopping time.

**Output:** Stopped predictor $T_{\hat{m}}$.

---

## 4.1. Anytime-Valid Conformal Prediction

Note that under a stopping rule, the calibration size $\hat{m}$ is a random variable that depends on the realized data. As a consequence, standard fixed-time concentration bounds do not apply at $\hat{m}$. In particular, a naive Hoeffding bonus of order $\sqrt{\log(2/\delta)/(2m)}$ is not valid under data-dependent stopping unless one applies a union bound over all $m$, which leads to overly conservative guarantees. To address this issue, we adopt an anytime-valid approach. Rather than relying on fixed-time concentration inequalities, we replace the standard Hoeffding bonus with a time-uniform confidence sequence. Specifically, we use the finite-LIL bounds of Howard et al. (2021), which hold simultaneously for all $m$ and therefore remain valid under arbitrary stopping rules. Let the time-uniform bonus be

$$r_m := 0.85 \sqrt{\frac{\log \log(2m) + 0.72 \log(10.4/\delta)}{m}}.$$

Given any stopping rule, we introduce the resulting time-uniform split conformal procedure in Algorithm 2. The following theorem establishes its validity.

**Theorem 4.1** (Anytime-valid split conformal under optional stopping). *Run Algorithm 2 with parameters $(\alpha, \delta)$ and an arbitrary stopping rule $\phi$. Then, with probability at least*

$1 - \delta$ *over the calibration stream* $(X_1, Y_1), \ldots, (X_n, Y_n)$, *we have*

$$\Pr_{(X,Y) \sim \mathcal{D}} \big(Y \notin T_{D_m}(X)\big) \leq \alpha \qquad \text{for all } m \in \{1, \ldots, n\}.$$

*In particular, the predictor returned at the stopping time $\hat{m}$ satisfies* $\Pr_{(X,Y) \sim \mathcal{D}} \big(Y \notin T_{D_{\hat{m}}}(X)\big) \leq \alpha$.

Theorem 4.1 first establishes validity uniformly over the entire sequence of predictors; the validity guarantee for the data-dependent stopping time $\hat{m}$ is then an immediate consequence of this uniform statement.

## 4.2. Design of the Stopping Rule

To introduce the stopping rule, we reparameterize the prediction sets in a way that simplifies the analysis of the cost function. Let $\widetilde{Y} \sim \mathrm{Unif}(\mathcal{Y})$ be independent of $(X, Y)$. Define the associated score random variables $S^\star := s(X, Y)$ and $S := s(X, \widetilde{Y})$, and let $F_S$ denote the CDF of $S$. For each $\beta \in [0, 1]$, define the threshold $\tau(\beta) := F_S^{-1}(1 - \beta)$, so that $\Pr(S > \tau(\beta)) = \beta$, and the corresponding prediction set

$$T_\beta(x) := T_{\tau(\beta)}(x) = \{y \in \mathcal{Y} : s(x, y) \leq \tau(\beta)\}.$$

This parameterization has the key advantage that the expected prediction-set size depends linearly on $\beta$:

$$\mathbb{E}_{\mathcal{D}}\big[\,|T_\beta(X)|\,\big] = \sum_{y \in \mathcal{Y}} \Pr_{\mathcal{D}}\big(s(X, y) \leq \tau(\beta)\big)$$
$$= K \Pr_{\mathcal{D}}\big(s(X, \widetilde{Y}) \leq \tau(\beta)\big) = K(1 - \beta).$$

Thus, $1 - \beta$ equals the expected prediction-set size normalized by $K$. It is therefore convenient to express miscoverage as a function of $\beta$.

**Definition 4.2** (Miscoverage curve and inverse). For $\beta \in [0, 1]$, define

$$L(\beta) := \Pr\big(Y \notin T_\beta(X)\big) = \Pr\big(S^\star > \tau(\beta)\big),$$

and its generalized inverse

$$L^{-1}(t) := \sup\{\beta \in [0, 1] : L(\beta) \leq t\}.$$

Given a calibration dataset $D_m = \{(X_i, Y_i)\}_{i=1}^{m}$, define the empirical miscoverage estimate

$$\widehat{L}_m(\beta) := \frac{1}{m} \sum_{i=1}^{m} \mathbf{1}\big\{s(X_i, Y_i) > \tau(\beta)\big\}.$$

Define the data-dependent size parameter

$$\beta_m := \frac{1}{nK} \sum_{i=1}^{n} \sum_{j=1}^{K} \mathbf{1}\big\{s(X_i, j) > \hat{q}_m\big\},$$

which can be interpreted as a noisy estimate of the oracle quantity $L^{-1}(\alpha - r_m)$. We are now ready to introduce our stopping rule. As we will show in Section 5, the stopping rule is based on computing a lower confidence bound on the first-order derivative of an *oracle* cost, where the empirical miscoverage estimate is replaced by its true expectation, formally defined in Definition 5.1. The procedure terminates when this lower confidence bound becomes positive, meaning that increasing the calibration set size further will incur additional cost.

**Definition 4.3.** For $m \geq 1$, define

$$r_m := 0.85 \sqrt{\frac{\log \log(2m) + 0.72 \log(10.4/\delta)}{m}},$$

$$t_m := \alpha - r_m,$$

$$\eta_m := 0.85 \sqrt{\frac{\log(1 + \log m) + \max\{7, \, 0.8 \log(1612n)\}}{m}}.$$

If $t_m \leq 0$ or $t_m + t_{m+1} \leq 0$, set $\underline{\Delta}_m := -\infty$. Otherwise, let

$$a_m := \frac{2\eta_m}{t_m} + \sqrt{\frac{\log(4n/\delta)}{n}},$$

$$u_m := \frac{2(r_m - r_{m+1})}{t_m + t_{m+1}},$$

$$\underline{\Delta}_m := (\beta_m - a_m) - (n - m - 1)\, u_m.$$

Define the stopping rule $\hat{\phi}$ as

$$\hat{\phi}_m = \begin{cases} 1 & \underline{\Delta}_m \geq 0 \\ 0 & \text{otherwise.} \end{cases}$$

We next state a monotonicity condition on the score distribution under which the above stopping rule is provably no-regret.

**Assumption 4.4** (Monotone density ratio for score distributions). Assume $S^\star$ and $S$ admit continuous densities $f_{S^\star}$ and $f_S$, with $f_S(\tau) > 0$ on the relevant range, and that the ratio

$$\tau \longmapsto \frac{f_{S^\star}(\tau)}{f_S(\tau)} \quad \text{is non-increasing.}$$

Empirical validation of Assumption 4.4 is provided in Appendix E. Assumption 4.4 formalizes the intuition that smaller scores are more likely under the true label than under a randomly chosen label. Under this assumption, the oracle cost, formally defined as $\bar{C}(m)$ in Definition 5.1 is convex. This convexity is a key structural property underpinning the design and analysis of our optimal stopping rule. Notably, the total expected cost $C$ need not be convex.

**Theorem 4.5** (Main Theorem). *Let $\hat{m}$ be the calibration set size induced by $\hat{\phi}$. When Assumption 4.4 holds, we have*

$$\mathbb{E}[C(\hat{m})] \leq C(m^\star) + R_n,$$

*where the regret term $R_n$ satisfies that $R_n = \tilde{O}(n^{2/3})$ and $\lim_{n \to \infty} \frac{R_n}{C(m^\star)} = 0$.*

We will formally define $R_n$ and explain the intuition behind the proof in Section 5. By combining Theorem 4.1 and Theorem 4.5, we conclude that applying Algorithm 2 with the stopping rule in Definition 4.3 guarantees both valid coverage and no-regret performance.

## 5. Analysis of the Stopping Rule

In this section, we analyze the stopping rule defined in Definition 4.3. We first study structural properties of an oracle notion of the cost function, showing that it is convex and hence unimodal under Assumption 4.4. This property provides useful global information about the cost landscape. We then derive the empirical stopping rule in Definition 4.3 by constructing data-dependent lower bounds on the increments of the oracle cost. Finally, we prove that the resulting data-dependent stopping time enjoys a no-regret guarantee, as stated in Theorem 4.5. The proofs in this section are deferred to Section A in the appendix.

### 5.1. Convexity of the Oracle Cost Function

For the analysis, we introduce two notions of cost. The first is the *realized* cost induced by the data-dependent size parameter $\beta_m$. The second is an *oracle cost* that replaces $\beta_m$ by its population counterpart $\bar{\beta}_m$; this cost will be convenient for establishing convexity.

**Definition 5.1** (Realized and oracle cost). Fix $m \in [n]$. Define the realized cost as

$$\widehat{C}(m) := m + (n - m)(1 - \beta_m),$$

and the oracle cost as

$$\overline{C}(m) := m + (n - m)(1 - \overline{\beta}_m),$$

where $\overline{\beta}_m := L^{-1}(\alpha - r_m)$.

From the definition it holds that the (true) expected cost at a fixed calibration size $m$ is

$$C(m) = \mathbb{E}\big[\widehat{C}(m)\big].$$

**Lemma 5.2** (Convexity of the loss function). *Under Assumption 4.4, the distributional loss function $L(\beta)$ is convex and continuous on $[0, 1]$.*

Intuitively, convexity follows from Assumption 4.4, which implies that the slope of $L$ is nondecreasing in $\beta$. Continuity follows from the continuity of the CDF $F_{S^\star}$. As a consequence of the convexity of $L$, we can establish a discrete convexity property for the oracle surrogate cost sequence.

**Lemma 5.3.** *The sequence $\{\overline{C}(m)\}_{m=1}^n$ is discrete convex under Assumption 4.4.*

The discrete convexity of $\overline{C}(m)$ is crucial for the design and analysis of the stopping rule. In particular, discrete convexity implies unimodality, which ensures that local comparisons of successive differences suffice to locate the global minimizer of the (oracle) cost sequence.

### 5.2. Derivation of the Stopping Rule

We now turn to the derivation of the empirical stopping rule. By Lemma 5.3, the oracle cost sequence $\{\overline{C}(m)\}_{m=1}^n$ is discrete convex and hence unimodal. It is well known that for a unimodal sequence $\{s_m\}_{m=1}^n$, a natural stopping rule is to stop at the first index $m$ such that $s_m - s_{m-1} \geq 0$. Accordingly, our goal is to identify the smallest $m$ such that the oracle cost increment $\Delta\overline{C}(m) := \overline{C}(m+1) - \overline{C}(m)$ becomes nonnegative. However, the oracle cost is defined in terms of population quantities and is not *directly observable*. We therefore construct a data-dependent lower bound on $\Delta\overline{C}(m)$ that can be estimated from data. We first show that the empirical size parameter $\beta_m$ concentrates uniformly around its oracle counterpart $\overline{\beta}_m$. Define the event

$$E_n := \bigcap_{m=1}^n \left\{ |\beta_m - \overline{\beta}_m| \leq a_m \right\}.$$

**Lemma 5.4** (Time-uniform quantile estimation). *For all $n$ large enough, $\Pr(E_n) \geq 1 - 1/n$.*

This result follows from time-uniform concentration bounds for the empirical CDF (Howard & Ramdas, 2022). Next, we control the increment of the oracle size parameter. Recall $u_m$ is defined in Definition 4.3.

**Lemma 5.5** (Upper bound of the difference). *For $m$ such that $t_m > 0$, $\overline{\beta}_{m+1} - \overline{\beta}_m \leq u_m$.*

Combining these results with the expression

$$\Delta\overline{C}(m) = \overline{\beta}_m - (n - m - 1)(\overline{\beta}_{m+1} - \overline{\beta}_m),$$

we define the data-dependent quantity

$$\underline{\Delta}_m := \left( \beta_m - \frac{2\eta_m}{t_m} - \sqrt{\frac{\log(4n/\delta)}{n}} \right) - (n-m-1)u_m.$$

By Lemmas 5.4 and 5.5, we have $\underline{\Delta}_m \leq \Delta\overline{C}(m)$ with high probability. The stopping condition $\underline{\Delta}_m \geq 0$ therefore serves as a data-dependent proxy for the oracle stopping rule and coincides with the definition given in Definition 4.3.

### 5.3. Cost comparison and no-regret guarantee

Let $\overline{m} := \inf \arg\min_{m \in [n]} \overline{C}(m)$ be the smallest oracle minimizer. We are now ready to present a proof sketch that explains the intuition behind Theorem 4.5.

*Proof sketch of Theorem 4.5.* We prove the theorem by first bounding the deviation $|\widehat{C}(m) - \overline{C}(m)|$ on the high-probability event $E_n$. We then show that the oracle cost difference $\overline{C}(\widehat{m}) - \overline{C}(\overline{m})$ can also be controlled, since the design of the stopping rule ensures that $\widehat{m}$ is neither too small nor too large. Combining these two bounds and taking expectations completes the proof. The full proof is deferred to Appendix A. $\square$

## 6. Connection to PAC Labeling

A recent work by Candès et al. (2025) introduces PAC (Probably Approximately Correct) labeling, a framework for constructing high-quality labeled datasets by trading off labeling accuracy against annotation cost. While the proposed method provides rigorous probabilistic guarantees on the labeling error, its treatment of labeling cost remains largely heuristic: reductions in annotation cost are demonstrated empirically, but without accompanying theoretical optimality guarantees.

In this section, we establish a formal connection between PAC labeling and conformal prediction. We first show that the PAC labeling framework of Candès et al. (2025) can be interpreted as a special case of binary-class conformal prediction, in which the PAC labeling error guarantee corresponds directly to a miscoverage guarantee. We then show that the problem of minimizing labeling cost in this setting admits a natural reduction to our cost-minimization formulation, thereby enabling a principled theoretical analysis of labeling efficiency.

**PAC Labeling.** We have a distribution $\mathcal{D}$ over $\mathcal{X} \times \mathcal{Y}$. Given an i.i.d. drawn unlabeled dataset $\{X_1, \cdots, X_n \in \mathcal{X}\}$, with *unknown* expert labels, our goal is to return a labeled dataset $\{(X_i, \tilde{Y}_i)\}_{i=1}^n$ such that with probability at least $1 - \delta$, we incur only a small amount of labeling errors:

$$\frac{1}{n} \sum_{i=1}^n \mathbf{1}[Y_i \neq \tilde{Y}_i] \leq \alpha. \tag{5}$$

Here $\alpha, \delta$ are user-chosen error parameters. To produce the label $\tilde{Y}_i$, we are allowed to query an expert for $Y_i$, which is costly or instead use a cheap AI prediction $\hat{Y}_i = f(X_i)$ where $f$ is a AI model. One can trivially achieve Equation (5) by collecting expert labels for all $n$ data points. However, the goal is to achieve the criterion while minimizing the cost of labeling, i.e. the number of expert queries. Candès et al. (2025) proposed a threshold-based method. First, they uniformly subsample $m$ data from unlabeled dataset $X_1, \cdots, X_n$, query expert for true labels and obtain a calibration dataset $D_m = \{(X_i, Y_i)\}_{i=1}^m$. Here we slightly abuse notation to use index from 1 to $m$ to represent calibration dataset. Then they use Algorithm 3 to learn a threshold $\hat{u}_m$ and query expert for the remaining data points whose uncertainties are above the learned threshold $\hat{u}_m$.

**Algorithm 3** Probably Approximately Correct Labeling

---

**Input:** unlabeled dataset $\{X_i\}_{i=1}^n$, calibration dataset $D_m = \{(X_i, Y_i)\}_{i=1}^m$, predictions $\{\hat{Y}_i\}_{i=1}^n$, uncertainties $\{U_i\}_{i=1}^n$, target labeling error $\alpha$, failure probability $\delta$.

1: Set the confidence bonus: $r_m = \sqrt{\frac{\log(2/\delta)}{2m}}$.
2: Let $\hat{u}_m$ be the smallest value such that

$$\frac{1}{m}\sum_{i=1}^m \mathbf{1}[Y_i \neq \hat{Y}_i] \cdot \mathbf{1}[U_i \leq \hat{u}_m] + r_m > \alpha \quad (6)$$

3: Let $\tilde{Y}_i = Y_i\mathbf{1}[U_i \geq \hat{u}_m] + \hat{Y}_i\mathbf{1}[U_i < \hat{u}_m]$.
**Output:** labeled dataset $\{(X_i, \tilde{Y}_i)\}_{i=1}^n$.

---

**Reduction to Binary-class Conformal Prediction.** Now we show that PAC labeling can be reduced to a binary-class conformal prediction. We can define a binary correctness label $A = \mathbf{1}[Y = \hat{Y}]$. And Let $Z = (X, \hat{Y}, U)$ be the observable features before expert querying. And we can define the non-conformity score function $s : Z \times A \to [0, 1]$ as $s(z, 1) = 0$, $s(z, 0) = 1 - u$. Then we can define the prediction set as

$$T_s(z, \hat{q}) := \{a \in \{0, 1\} : s(z, a) \leq \hat{q}\}.$$

Or equivalently, let $\hat{u} = 1 - \hat{q}$, we obtain the two-valued prediction set family

$$T_{\hat{u}}(z) = \begin{cases} \{1\}, & u < \hat{u}, \\ \{0, 1\}, & u \geq \hat{u}, \end{cases}$$

Under this prediction set, we have $\mathbf{1}[a \notin T_s(z, \tau)] = \mathbf{1}[u \leq \hat{u}] \cdot \mathbf{1}[a = 0]$. So, the miscoverage in conformal prediction happens if the cheap label in the pac is wrong and the algorithm decides not to ask for a human label. Formally we have

$$\frac{1}{m}\sum_{i=1}^m \mathbf{1}[Y_i \neq \hat{Y}_i] \cdot \mathbf{1}[U_i \leq \hat{u}_m] = \frac{1}{m}\sum_{i=1}^m \mathbf{1}\{A_i \notin T_{\hat{u}}(x_i)\}.$$

**By rearranging the equality, we observe that in the PAC labeling setting, solving Equation (6) is equivalent to solving Equation (1).** Consequently, their method can be interpreted as an instance of conformal prediction. Last, we will show the expected number of expert queries, denoted as $Q(m)$, can be reduced to our cost formulation Equation (2).

$$\begin{aligned} Q(m) &:= m + (n - m)(1 - \mathbb{E}[\mathbf{1}[u \leq \hat{u}_m]]) \\ &= m + (n - m)(1 - \mathbb{E}[\mathbf{1}[s(z, 0) > \hat{q}_m]]) \\ &= m + (n - m)\mathbb{E}[s(z, 0) \leq \hat{q}_m] \\ &= m + (n - m)\mathbb{E}[\text{len}(T_s(x, \hat{q}_m) - 1] \\ &= 2m - n + \mathbb{E}[\text{len}(T_s(x, \hat{q}_m)]. \end{aligned}$$

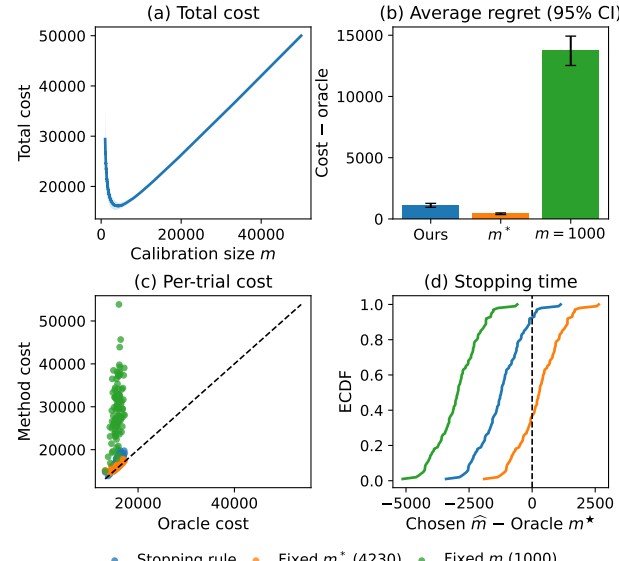

*Figure 1.* Conformal prediction with costly calibration on ImageNet. (a) Total cost versus calibration size. (b) Average regret relative to the oracle. (c) Per-trial method cost versus oracle cost. (d) ECDF of $\hat{m} - m^\star$. We compare the proposed stopping rule to a fixed calibration size $m = 1000$ commonly used in ImageNet conformal prediction (Angelopoulos & Bates, 2021).

Since $n$ is fixed, minimizing $Q(m)$ is equivalent to minimizing the cost of the corresponding conformal prediction defined as $C(m) = m + \mathbb{E}[\text{len}(T_s(x, \hat{q}_m)/K]$ where $K = 2$ since $A$ is binary label. In light of the above reduction, Assumption 4.4 can be equivalently expressed as the following condition in the PAC labeling setting.

**Assumption 6.1.** $u \mapsto \mathbb{E}[\mathbf{1}[Y \neq \hat{Y}] \mid U = u]$ is increasing.

Empirical validation of Assumption 6.1 across datasets is provided in Appendix E. Intuitively, this assumption states that predictions with higher uncertainty are more likely to be incorrect. Under the above assumption, we obtain an analogous no-regret guarantee for the labeling cost.

**Theorem 6.2.** *Let $\hat{m}$ be the calibration set size induced by $\hat{\phi}$. When Assumption 6.1 holds, we have*

$$\mathbb{E}[Q(\hat{m})] \leq Q(m^\star) + R_n,$$

*where the regret term $R_n$ satisfies that $R_n = \tilde{O}(n^{2/3})$ and $\lim_{n\to\infty} \frac{R_n}{Q(m^\star)} = 0$.*

To prove this theorem, we first establish the equivalence between Assumption 4.4 and Assumption 6.1 in the PAC labeling setting. The result then follows immediately by applying Theorem 4.5. The detailed proof appears in Appendix C.

# 7. Experiments

We empirically evaluate the proposed stopping rule in settings where prediction sets are constructed via split conformal prediction and calibration labels are costly. We show that the stopping rule selects calibration sizes close to the oracle optimum in terms of the total cost defined in Section 4, while maintaining valid coverage guarantees, in both conformal prediction and PAC labeling settings.

**Finite-sample stabilization.** The stopping rule in Section 4 is derived under asymptotic assumptions. In finite samples, we introduce a stabilization term by replacing $\alpha - r_m$ with

$$t_m := \frac{\kappa}{\sqrt{n}} + \alpha - r_m, \qquad \underline{\Delta}_m := \beta_m - \frac{2\eta_m}{t_m}.$$

Here $\kappa > 0$ is a finite-sample hyperparameter that controls the aggressiveness of stopping. This ensures that the stopping condition is attainable in moderate sample sizes, where the time-uniform radius $r_m$ may otherwise prevent stopping. The additional term $\kappa/\sqrt{n}$ vanishes as $n \to \infty$, so $t_m$ converges to the asymptotic quantity $\alpha - r_m$ used in the theoretical analysis.

## 7.1. Optimal Stopping in Conformal Prediction

**Dataset and Setting.** We evaluate on the ImageNet validation set using a pretrained ResNet-152 classifier (He et al., 2016). We use the nonconformity score $s(x, y) = 1 - \hat{p}(y \mid x)$. For each calibration size $m$, we construct an anytime-valid split conformal predictor with miscoverage level $\alpha = 0.1$ and failure probability $\delta = 0.05$ using Algorithm 2 with finite-sample stabilization, where $\kappa = 20$. Results are averaged over 100 trials. We compare against a fixed calibration size $m = 1000$, commonly used in ImageNet conformal prediction (Angelopoulos & Bates, 2021), and the best fixed calibration size chosen in hindsight.

**Conformal Prediction Results.** Figure 1(a) plots the realized total cost as a function of the calibration size and exhibits a clear convex shape with a minimum at $m^* = 4230$. Figure 1(b) reports average regret relative to this oracle fixed calibration size, where the stopping rule substantially outperforms the naive choice $m = 1000$ and performs comparably to the best fixed $m^*$. Figure 1(c) compares per-trial costs against a per-trial oracle; the stopping rule closely tracks the oracle, while the fixed baseline incurs consistently higher cost. Overall, the stopping rule achieves a $40.6\% \pm 2.3\%$ (95% CI) reduction in total cost relative to the baseline. Finally, Figure 1(d) shows the empirical CDF of the difference between the chosen and oracle calibration sizes, indicating that the stopping rule typically stops near the oracle optimum, whereas the naive baseline stops too early and never reaches the oracle calibration size.

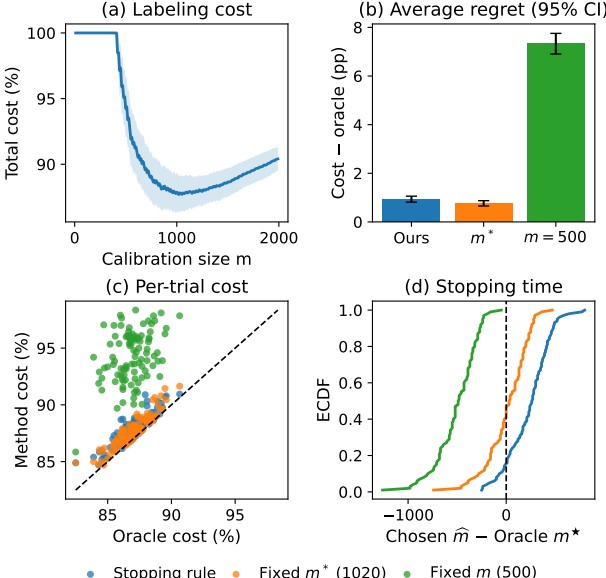

*Figure 2.* PAC labeling on the media bias dataset. (a) Labeling cost versus calibration size, measured as the percentage of expert-labeled data used. (b) Average regret relative to the oracle, measured in percentage points (pp). (c) Per-trial labeling cost versus oracle cost. (d) ECDF of $\hat{m} - m^*$. We compare the stopping rule to the fixed calibration size $m = 500$ used in prior PAC labeling work (Candès et al., 2025).

## 7.2. Optimal Stopping in PAC Labeling

We now present experimental results for the PAC labeling setting described in Section 6.

**Datasets and Settings.** We evaluate on political bias labels for media articles (Baly et al., 2020), as used in prior PAC labeling work (Candès et al., 2025), where each example $(X_i, Y_i)$ is a media article labeled by political orientation with $Y_i \in \mathcal{Y} = \{\text{left}, \text{center}, \text{right}\}$. Predicted labels $\hat{Y}_i$ are obtained from GPT-4o, following the setup of (Gligorić et al., 2025). For the uncertainty scores $U_i$, we use GPT's verbalized confidence estimates as proposed in (Tian et al., 2023). We use zero–one loss and follow the experimental setup of the original PAC labeling baseline, with $\alpha = 0.1$ and $\delta = 0.05$. We set $\kappa = 30$ and average results over 100 trials. Additional results on ImageNet (He et al., 2016), misinformation detection (Gabriel et al., 2022), and media stance on global warming (Luo et al., 2020), together with comparisons against a practical greedy stopping heuristic, are provided in the Appendix F. As a baseline, we compare against the fixed calibration size used in the original PAC labeling method, as well as the oracle calibration size chosen in hindsight.

**PAC Labeling Results.** Figure 2 reports results on the media bias dataset, where performance is measured by the percentage of expert labeling cost required to satisfy the PAC

guarantee. Our stopping rule again *consistently achieves substantial savings* in human labeling cost relative to the fixed calibration baseline used in prior PAC labeling work, while closely matching the oracle calibration size chosen in hindsight. On the media bias dataset, the stopping rule reduces labeling cost by $6.4\% \pm 0.5\%$ (95% CI) relative to the baseline. In particular, the stopping rule selects calibration sizes near-optimal for minimizing total labeling budget, whereas the naive fixed baseline stops too early and never reaches the oracle calibration size. Consistent trends are observed across additional datasets, which we report in Appendix F.

## Acknowledgements

ZSW was supported in part by NSF Awards 2339775 and 2232693.

## Impact Statement

This paper presents work whose goal is to advance the field of machine learning. There are many potential societal consequences of our work, none of which we feel must be specifically highlighted here.

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

# A. Proof in Section 4 and Section 5

**Theorem 4.1** (Anytime-valid split conformal under optional stopping). *Run Algorithm 2 with parameters $(\alpha, \delta)$ and an arbitrary stopping rule $\phi$. Then, with probability at least $1 - \delta$ over the calibration stream $(X_1, Y_1), \ldots, (X_n, Y_n)$, we have*

$$\Pr_{(X,Y) \sim \mathcal{D}} \big( Y \notin T_{D_m}(X) \big) \leq \alpha \qquad \text{for all } m \in \{1, \ldots, n\}.$$

*In particular, the predictor returned at the stopping time $\hat{m}$ satisfies $\Pr_{(X,Y) \sim \mathcal{D}} \big( Y \notin T_{D_{\hat{m}}}(X) \big) \leq \alpha$.*

*Proof.* Let $S_i^\star := s(X_i, Y_i)$ and let $F^\star$ denote the CDF of $S^\star := s(X, Y)$ under $(X, Y) \sim \mathcal{D}$. Let $\widehat{F}_m$ be the empirical CDF of $S_1^\star, \ldots, S_m^\star$:

$$\widehat{F}_m(q) := \frac{1}{m} \sum_{i=1}^{m} \mathbf{1}[S_i^\star \leq q].$$

Recalling

$$r_m = 0.85 \sqrt{\frac{\log\log(2m) + 0.72 \log(10.4/\delta)}{m}}.$$

Let

$$G := \Big\{ \forall m \in \{1, \ldots, n\} : \|\widehat{F}_m - F^\star\|_\infty \leq r_m \Big\}.$$

By the time-uniform finite-LIL DKW bound Howard & Ramdas (2022, Thm. 2), for any $C \geq 7$,

$$\Pr \left( \exists m \geq 1 : \|\widehat{F}_m - F^\star\|_\infty > 0.85 \sqrt{\frac{\log\log(em) + C}{m}} \right) \leq 1612 e^{-1.25C}.$$

We have that $\Pr(G) \geq 1 - \delta$.

Work on $G$ and fix any $m \in \{1, \ldots, n\}$. Let $u_m := 1 - \alpha + r_m$. If $u_m \geq 1$, Algorithm 2 sets $\hat{q}_m = +\infty$, so $T_m(x) = \mathcal{Y}$ and $\Pr(Y \notin T_m(X)) = 0 \leq \alpha$.

Otherwise $u_m < 1$ and $\hat{q}_m$ is chosen so that $\widehat{F}_m(\hat{q}_m) \geq u_m$. On $G$, $F^\star(\hat{q}_m) \geq \widehat{F}_m(\hat{q}_m) - r_m \geq (1 - \alpha + r_m) - r_m = 1 - \alpha$. Therefore

$$\Pr_{(X,Y) \sim \mathcal{D}} \big( Y \notin T_m(X) \big) = \Pr \big( S^\star > \hat{q}_m \big) = 1 - F^\star(\hat{q}_m) \leq \alpha.$$

Since the same event $G$ implies the bound for every $m \in \{1, \ldots, n\}$, it also holds at the stopping time $\hat{m}$ (induced by any stopping rule $\phi$):

$$\Pr_{(X,Y) \sim \mathcal{D}} \big( Y \notin T_{D_{\hat{m}}}(X) \big) \leq \alpha.$$

$\square$

Now we turn to prove Theorem 4.5. We begin with proving the continuity of the loss curve, which is very useful in the analysis.

**Lemma A.1** (Continuity of the inverse). *For any $t > 0$ and any $\eta \in (0, t)$, $L^{-1}(t + \eta) - L^{-1}(t - \eta) \leq \frac{2\eta}{t}$.*

*Proof.* Let $\beta_t := L^{-1}(t) = \sup\{\beta \in [0, 1] : L(\beta) \leq t\}$. Since $L$ is convex with $L(0) = 0$, the map $\beta \mapsto L(\beta)/\beta$ is nondecreasing on $(0, 1]$. Moreover, since $L$ is continuous and nondecreasing, we have $L(\beta_t) = t$ (for $t$ in the range of $L$ and $\beta_t < 1$).

For any $\beta \geq \beta_t$,

$$\frac{L(\beta)}{\beta} \geq \frac{L(\beta_t)}{\beta_t} = \frac{t}{\beta_t}$$

Therefore, $L(\beta) \geq \beta \cdot \frac{t}{\beta_t}$.

Hence if $\beta > \beta_t \cdot \frac{t+\eta}{t} = \beta_t \left( 1 + \frac{\eta}{t} \right)$, then $L(\beta) > t + \eta$, so $L^{-1}(t + \eta) \leq \beta_t \left( 1 + \frac{\eta}{t} \right)$.

Similarly, for any $\beta \leq \beta_t$,

$$\frac{L(\beta)}{\beta} \leq \frac{L(\beta_t)}{\beta_t} = \frac{t}{\beta_t}.$$

Therefore, $L(\beta) \leq \beta \cdot \frac{t}{\beta_t}$.

Thus if $\beta < \beta_t \cdot \frac{t - \eta}{t} = \beta_t \left(1 - \frac{\eta}{t}\right)$, then $L(\beta) < t - \eta$, so $L^{-1}(t - \eta) \geq \beta_t \left(1 - \frac{\eta}{t}\right)$.

Combining and using $\beta_t \leq 1$, we have $L^{-1}(t + \eta) - L^{-1}(t - \eta) \leq \beta_t \left(\frac{2\eta}{t}\right) \leq \frac{2\eta}{t}$. $\qquad\square$

**Lemma 5.2** (Convexity of the loss function). *Under Assumption 4.4, the distributional loss function $L(\beta)$ is convex and continuous on $[0, 1]$.*

*Proof.* Define the survival functions $\bar{F}_S(t) := \Pr(S > t)$ and $\bar{F}_{S^\star}(t) := \Pr(S^\star > t)$, so that $\beta = \bar{F}_S(\tau(\beta))$, $L(\beta) = \bar{F}_{S^\star}(\tau(\beta))$. Since $F_S$ is continuous and strictly increasing under Assumption 4.4, the quantile map $\tau(\beta) = F_S^{-1}(1 - \beta)$ is well-defined, continuous, and strictly decreasing in $\beta$.

Let $\lambda(t) := \frac{f_{S^\star}(t)}{f_S(t)}$, which is nonincreasing by Assumption 4.4. Fix $0 < \beta_1 < \beta_2 < 1$ and write $t_i := \tau(\beta_i)$, so $t_1 > t_2$. Using that $S$ and $S^\star$ admit densities, we have

$$\beta_2 - \beta_1 = \bar{F}_S(t_2) - \bar{F}_S(t_1) = \int_{t_2}^{t_1} f_S(u)\, du, \qquad L(\beta_2) - L(\beta_1) = \bar{F}_{S^\star}(t_2) - \bar{F}_{S^\star}(t_1) = \int_{t_2}^{t_1} f_{S^\star}(u)\, du.$$

Therefore the secant slope can be written as a weighted average of $r$:

$$\frac{L(\beta_2) - L(\beta_1)}{\beta_2 - \beta_1} = \frac{\int_{t_2}^{t_1} f_{S^\star}(u)\, du}{\int_{t_2}^{t_1} f_S(u)\, du} = \frac{\int_{t_2}^{t_1} \lambda(u)\, f_S(u)\, du}{\int_{t_2}^{t_1} f_S(u)\, du},$$

where the denominator is positive since $f_S > 0$ on the relevant range.

Now take $0 < \beta_1 < \beta_2 < \beta_3 < 1$ and set $t_i = \tau(\beta_i)$, so $t_1 > t_2 > t_3$. Let

$$s_{12} := \frac{L(\beta_2) - L(\beta_1)}{\beta_2 - \beta_1}, \qquad s_{23} := \frac{L(\beta_3) - L(\beta_2)}{\beta_3 - \beta_2}.$$

Because $r$ is nonincreasing, for all $u \in [t_2, t_1]$ we have $r(u) \leq r(t_2)$, hence

$$s_{12} = \frac{\int_{t_2}^{t_1} \lambda(u)\, f_S(u)\, du}{\int_{t_2}^{t_1} f_S(u)\, du} \leq r(t_2).$$

Similarly, for all $u \in [t_3, t_2]$ we have $r(u) \geq r(t_2)$, hence

$$s_{23} = \frac{\int_{t_3}^{t_2} \lambda(u)\, f_S(u)\, du}{\int_{t_3}^{t_2} f_S(u)\, du} \geq r(t_2).$$

Combining yields $s_{12} \leq s_{23}$, i.e., the secant slopes of $L$ are nondecreasing. This is equivalent to convexity of $L$ on $(0, 1)$ (and hence on $[0, 1]$).

Finally, $F_{S^\star}$ is continuous and $\tau(\beta)$ is continuous, so $L(\beta) = 1 - F_{S^\star}(\tau(\beta))$ is continuous in $\beta$ on $[0, 1]$. This completes the proof. $\qquad\square$

**Lemma 5.3.** *The sequence $\{\overline{C}(m)\}_{m=1}^n$ is discrete convex under Assumption 4.4.*

*Proof.* By Lemma 5.2, $L$ is convex, continuous, and nondecreasing on $[0, 1]$. We first note that its generalized inverse

$$L^{-1}(t) := \sup\{\beta \in [0, 1] : L(\beta) \leq t\}$$

is concave and nondecreasing in $t$. Indeed, monotonicity is immediate since the feasible set $\{\beta : L(\beta) \le t\}$ expands with $t$. For concavity, fix $t_1, t_2 \in [0, 1]$ and $\lambda \in [0, 1]$, and set $\beta_i := L^{-1}(t_i)$. By continuity and monotonicity of $L$, we have $L(\beta_i) \le t_i$ for $i = 1, 2$. Then by convexity of $L$,

$$L\big(\lambda\beta_1 + (1 - \lambda)\beta_2\big) \le \lambda L(\beta_1) + (1 - \lambda)L(\beta_2) \le \lambda t_1 + (1 - \lambda)t_2,$$

so $\lambda\beta_1 + (1 - \lambda)\beta_2$ is feasible for $L(\beta) \le \lambda t_1 + (1 - \lambda)t_2$. Taking the supremum over feasible $\beta$ yields

$$L^{-1}\big(\lambda t_1 + (1 - \lambda)t_2\big) \ge \lambda L^{-1}(t_1) + (1 - \lambda)L^{-1}(t_2),$$

so $L^{-1}$ is concave.

Next, recall

$$r_m = 0.85\sqrt{\frac{\log\log(2m) + 0.72\log(10.4/\delta)}{m}},$$

Taking the derivative over $m$, it can be verified that $r_m$ is decreasing and convex for $m \ge 2$, which implies that $t_m$ is increasing and concave.

Define $\overline{\beta}_m := L^{-1}(t_m)$. Since $L^{-1}$ is concave and nondecreasing and $(t_m)$ is discrete concave, for $m = 2, \ldots, n-1$ we have

$$\overline{\beta}_m = L^{-1}(t_m) \ge L^{-1}\left(\frac{t_{m-1} + t_{m+1}}{2}\right) \ge \frac{L^{-1}(t_{m-1}) + L^{-1}(t_{m+1})}{2} = \frac{\overline{\beta}_{m-1} + \overline{\beta}_{m+1}}{2}.$$

Thus $(\overline{\beta}_m)$ is discrete concave. Since $t_m$ is increasing and $L^{-1}$ is nondecreasing, $(\overline{\beta}_m)$ is also nondecreasing.

Let $\delta_m := \overline{\beta}_{m+1} - \overline{\beta}_m$ for $m = 1, \ldots, n-1$. Because $(\overline{\beta}_m)$ is nondecreasing, $\delta_m \ge 0$, and because it is discrete concave, $(\delta_m)$ is nonincreasing.

Compute the first difference:

$$\Delta\overline{C}(m) := \overline{C}(m+1) - \overline{C}(m)$$
$$= \overline{\beta}_m - (n - m - 1)(\overline{\beta}_{m+1} - \overline{\beta}_m),.$$

Therefore, for $m = 1, \ldots, n-2$,

$$\Delta\overline{C}(m+1) - \Delta\overline{C}(m) = \overline{\beta}_{m+1} - (n - m - 2)\delta_{m+1} - \overline{\beta}_m + (n - m - 1)\delta_m$$
$$= (n - m)\delta_m - (n - m - 2)\delta_{m+1}$$
$$= (n - m)(\delta_m - \delta_{m+1}) + 2\delta_{m+1} \ge 0,$$

since $\delta_m \ge \delta_{m+1}$ and $\delta_{m+1} \ge 0$. Hence $\Delta\overline{C}(m)$ is nondecreasing, i.e. $(\overline{C}(m))_{m=1}^n$ is discrete convex. $\qquad\square$

**Lemma 5.4** (Time-uniform quantile estimation). *For all $n$ large enough, $\Pr(E_n) \ge 1 - 1/n$.*

*Proof.* By sampling with replacement, $\{s_i^\star\}_i$ are i.i.d. with CDF $F_{S^\star}$. Let $\widehat{F}_m$ be the empirical CDF of $\{s_i^\star\}_i$.

By the empirical-process finite-LIL bound (Howard & Ramdas, 2022), we have that with probability at least $1 - \frac{1}{2n}$ we have for all $m$,

$$\|\widehat{F}_m - F_{S^\star}\|_\infty \le \eta_m.$$

Fix such an $m \ge m_0$. For any $\beta \in [0, 1]$,

$$\widehat{L}_m(\beta) = \frac{1}{m}\sum_{j=1}^m \mathbf{1}\{s_j^\star > \tau(\beta)\} = 1 - \widehat{F}_m(\tau(\beta)), \qquad L(\beta) = \Pr(s_1^\star > \tau(\beta)) = 1 - F_{S^\star}(\tau(\beta)).$$

Therefore,

$$\sup_{\beta\in[0,1]} |\widehat{L}_m(\beta) - L(\beta)| \le \|\widehat{F}_m - F_{S^\star}\|_\infty \le \eta_m.$$

Define $\beta'_m := \sup\{\beta : \widehat{L}_m(\beta) \le t_m\}$ and $\overline{\beta}_m := L^{-1}(t_m)$. If $L(\beta) \le t_m - \eta_m$ then $\widehat{L}_m(\beta) \le t_m$, so $\beta$ is feasible and $\beta'_m \ge L^{-1}(t_m - \eta_m)$. If $L(\beta) > t_m + \eta_m$ then $\widehat{L}_m(\beta) > t_m$, so $\beta$ is infeasible and $\beta'_m \le L^{-1}(t_m + \eta_m)$. Hence

$$L^{-1}(t_m - \eta_m) \le \beta_m \le L^{-1}(t_m + \eta_m).$$

For all large $n$, since $m_0 \to \infty$ and $r_m \to 0$, we have $t_m \ge \alpha/2$ for all $m \ge m_0$, and also $\eta_m < t_m$. Applying Lemma A.1 gives

$$|\beta'_m - \overline{\beta}_m| \le L^{-1}(t_m + \eta_m) - L^{-1}(t_m - \eta_m) \le \frac{2\eta_m}{t_m}.$$

Next, we bound the difference between $\beta_m$ and $\beta'_m$ for all $m$. From the definition of $\beta_m$, this hold by directly applying the DKW inequality for a single size $n$ and budget $\frac{1}{2n}$.

Therefore, with probability $1 - \frac{1}{2n}$, for any $m$, we have

$$|\beta'_m - \beta_m| \le \sqrt{\frac{\log(4n/\delta)}{n}}.$$

Intersecting over $m$ and taking a union bound complete the proof. $\qquad\square$

**Lemma 5.5** (Upper bound of the difference). *For $m$ such that $t_m > 0$, $\overline{\beta}_{m+1} - \overline{\beta}_m \le u_m$.*

*Proof.* Let $\eta := t_{m+1} - t_m = r_m - r_{m+1} > 0$. Apply Lemma A.1 with $t = (t_m + t_{m+1})/2 = t_m + \eta/2$ and $\eta' = \eta/2$, so that $t + \eta' = t_{m+1}$ and $t - \eta' = t_m$. Then

$$\overline{\beta}_{m+1} - \overline{\beta}_m = L^{-1}(t_{m+1}) - L^{-1}(t_m) \le \frac{2\eta'}{t} = \frac{\eta}{t_m + \eta/2} = \frac{2(r_m - r_{m+1})}{t_m + t_{m+1}}.$$

$\qquad\square$

Define $m_0 := \left\lceil n^{2/3}/\log(n) \right\rceil$.

**Lemma A.2.** *There exists $n_0$ such that for all $n \ge n_0$, we have $\overline{m} \ge m_0$.*

*Proof.* It suffices to show $\Delta\overline{C}(m_0) < 0$ for all large $n$. Recall that

$$\overline{C}(m) = m + (n - m)(1 - \overline{\beta}_m), \qquad \overline{\beta}_m := L^{-1}(t_m), \quad t_m = \alpha - r_m,$$

so

$$\Delta\overline{C}(m) = \overline{C}(m+1) - \overline{C}(m) = \overline{\beta}_m - (n - m - 1)(\overline{\beta}_{m+1} - \overline{\beta}_m).$$

Using $\overline{\beta}_{m_0} \le 1$ gives

$$\Delta\overline{C}(m_0) \le 1 - (n - m_0 - 1)(\overline{\beta}_{m_0+1} - \overline{\beta}_{m_0}).$$

We first lower bound $\overline{\beta}_{m_0+1} - \overline{\beta}_{m_0}$ from convexity of $L$. Let $\beta^\star := L^{-1}(\alpha) \in (0, 1)$. Since $t_m = \alpha - r_m < \alpha$, monotonicity of $L^{-1}$ yields $\overline{\beta}_m \le \beta^\star$ for all $m$. Define $M_\star := \frac{1-\alpha}{1-\beta^\star}$. By convexity of $L$ and $L(1) = 1$, the function $L$ is $M_\star$-Lipschitz on $[0, \beta^\star]$, i.e. for all $0 \le \beta_1 \le \beta_2 \le \beta^\star$,

$$L(\beta_2) - L(\beta_1) \le M_\star(\beta_2 - \beta_1).$$

Apply this with $\beta_1 = \overline{\beta}_{m_0}$ and $\beta_2 = \overline{\beta}_{m_0+1}$ to get

$$t_{m_0+1} - t_{m_0} = L(\overline{\beta}_{m_0+1}) - L(\overline{\beta}_{m_0}) \le M_\star(\overline{\beta}_{m_0+1} - \overline{\beta}_{m_0}).$$

Since $t_{m_0+1} - t_{m_0} = r_{m_0} - r_{m_0+1}$, we obtain

$$\overline{\beta}_{m_0+1} - \overline{\beta}_{m_0} \ge \frac{r_{m_0} - r_{m_0+1}}{M_\star}.$$

Plugging into the bound on $\Delta \overline{C}(m_0)$ yields

$$\Delta \overline{C}(m_0) \leq 1 - \frac{n - m_0 - 1}{M_\star}(r_{m_0} - r_{m_0+1}).$$

We have $r_{m_0} - r_{m_0+1} = \Omega(m_0^{-3/2})$ (up to a log term), so

$$(n - m_0 - 1)(r_{m_0} - r_{m_0+1}) \to \infty.$$

Since $M_\star$ is a fixed (distribution-dependent) constant, the right-hand side becomes negative for all large $n$. This implies $\Delta \overline{C}(m_0) < 0$ and therefore $\overline{m} \geq m_0$ for all large $n$. $\qquad\square$

**Lemma A.3** (No Earlier Stopping). *On $E_n$, if $\overline{m} \leq n - 1$ then $\widehat{m} - 1 \geq \overline{m}$, and if $\overline{m} = n$ then $\widehat{m} = n$.*

*Proof.* On $E_n$, $\overline{\beta}_m \geq \beta_m - a_m$ for all $m \geq m_0$. Also, by Lemma 5.5, $\overline{\beta}_{m+1} - \overline{\beta}_m \leq u_m$. Thus on $E_n$,

$$\overline{\beta}_m - (n - m - 1)(\overline{\beta}_{m+1} - \overline{\beta}_m) \geq (\beta_m - a_m) - (n - m - 1)u_m.$$

Hence on $E_n$, $\underline{\Delta}_m \geq 0 \;\Rightarrow\; \Delta \overline{C}(m) \geq 0$.

Since $\overline{C}$ is discrete convex (Lemma 5.3), $m \mapsto \Delta \overline{C}(m)$ is nondecreasing. If $\overline{m} \leq n - 1$, then the first index where $\Delta \overline{C}(m) \geq 0$ is exactly $\overline{m}$, so the first $m$ with $\underline{\Delta}_m \geq 0$ must satisfy $m \geq \overline{m}$, i.e. $\widehat{m} - 1 \geq \overline{m}$. If $\overline{m} = n$, then $\Delta \overline{C}(m) < 0$ for all $m \leq n - 1$, hence $\underline{\Delta}_m < 0$ for all $m \leq n - 1$ on $E_n$, so the set $\{m : \underline{\Delta}_m \geq 0\}$ is empty and $\widehat{m} = n$. $\qquad\square$

Let $\beta^\star := L^{-1}(\alpha)$. From the fact that $L(0) = 0, L(1) = 1$ and that $L(\beta)$ being continuous, we know that $\beta^\star \in (0,1)$. Define $B_\delta := 0.72 \log(10.4/\delta), c_n := 0.85\sqrt{\log\log(2n) + B_\delta}$. Recall the time-uniform mean-CS radius satisfies for all $m \leq n$,

$$r_m = 0.85\sqrt{\frac{\log\log(2m) + B_\delta}{m}} \leq \frac{c_n}{\sqrt{m}}.$$

**Theorem 4.5** (Main Theorem). *Let $\hat{m}$ be the calibration set size induced by $\hat{\phi}$. When Assumption 4.4 holds, we have*

$$\mathbb{E}[C(\hat{m})] \leq C(m^\star) + R_n,$$

*where the regret term $R_n$ satisfies that $R_n = \tilde{O}(n^{2/3})$ and $\lim_{n\to\infty} \frac{R_n}{C(m^\star)} = 0$.*

*Proof.* Define $M_n := \left\lceil \left(\frac{32nc_n}{\alpha\beta^\star}\right)^{2/3} \right\rceil, R_n = M_n + b_n + 1, b_n = \max_{m_0 \leq m \leq n}(n - m)a_m, m_0 = \left\lceil n^{2/3}/\log(n) \right\rceil$. We start the proof with proving the regret bound.

**Step 1: Bound the stopping index.** We work on the high probability event $E_n$ and fix $m \geq \max\{m_0, M_n\}$.

For all large $n$, since $m_0 \to \infty$ and $r_m \to 0$, we have $t_m = \alpha - r_m \geq \alpha/2$ for all $m \geq m_0$, and also $a_m \leq a_{m_0} \to 0$. In particular, for all large $n$ we may assume

$$a_{m_0} \leq \frac{\beta^\star}{32}, \qquad \frac{2r_{m_0}}{\alpha} \leq \frac{\beta^\star}{32}. \tag{7}$$

Next, we bound $\beta^\star - \overline{\beta}_m$ using Lemma A.1. Let $t = \alpha - r_m/2$ and $\eta = r_m/2$ so that $t + \eta = \alpha$ and $t - \eta = \alpha - r_m = t_m$. Then

$$\beta^\star - \overline{\beta}_m = L^{-1}(\alpha) - L^{-1}(\alpha - r_m) \leq \frac{r_m}{\alpha - r_m/2} \leq \frac{2r_m}{\alpha} \leq \frac{2r_{m_0}}{\alpha} \leq \frac{\beta^\star}{32},$$

where we used $r_m$ is nonincreasing and equation 7. Hence

$$\overline{\beta}_m \geq \beta^\star - \frac{\beta^\star}{32}. \tag{8}$$

Now use $E_n$: since $|\beta_m - \overline{\beta}_m| \le a_m$ on $E_n$,

$$\beta_m - a_m \ge \overline{\beta}_m - 2a_m \ge \overline{\beta}_m - 2a_{m_0} \ge \left(\beta^\star - \frac{\beta^\star}{32}\right) - 2 \cdot \frac{\beta^\star}{32} = \frac{29\beta^\star}{32},$$

using equation 8 and equation 7.

It remains to control the $u_m$ term. Write $r_m = c_m/\sqrt{m}$ with

$$c_m := 0.85\sqrt{\log\log(2m) + B_\delta},$$

so $c_m$ is nondecreasing and $c_m \le c_n$ for $m \le n$. Then

$$r_m - r_{m+1} = \frac{c_m}{\sqrt{m}} - \frac{c_{m+1}}{\sqrt{m+1}} \le c_{m+1}\left(\frac{1}{\sqrt{m}} - \frac{1}{\sqrt{m+1}}\right) \le c_n\left(\frac{1}{\sqrt{m}} - \frac{1}{\sqrt{m+1}}\right) \le \frac{c_n}{2m^{3/2}},$$

where we used $c_m \le c_{m+1}$ and the standard inequality $\frac{1}{\sqrt{m}} - \frac{1}{\sqrt{m+1}} \le \frac{1}{2m^{3/2}}$.

Since $t_m \ge \alpha/2$ for $m \ge m_0$,

$$(n - m - 1)u_m = (n - m - 1)\frac{r_m - r_{m+1}}{t_m} \le \frac{n(r_m - r_{m+1})}{t_m} \le \frac{nc_n}{m^{3/2}\alpha}.$$

By the definition of $M_n$, for $m \ge M_n$ we have $m^{3/2} \ge 32nc_n/(\alpha\beta^\star)$, hence

$$(n - m - 1)u_m \le \frac{\beta^\star}{32}.$$

Combining the bounds gives, on $E_n$,

$$\underline{\Delta}_m = (\beta_m - a_m) - (n - m - 1)u_m \ge \frac{29\beta^\star}{32} - \frac{\beta^\star}{32} = \frac{7\beta^\star}{8} > 0.$$

Therefore the stopping condition must be met by $m = \max\{m_0, M_n\}$ on $E_n$.

**Step 2: Bound $|\widehat{C}(m) - \overline{C}(m)|$.**   On $E$, for all $m \ge m_0$,

$$|\widehat{C}(m) - \overline{C}(m)| = (n - m)|\beta_m - \overline{\beta}_m| \le (n - m)a_m \le b_n.$$

**Step 3: Bound $\overline{C}(\widehat{m}) - \overline{C}(\overline{m})$.**   If $\overline{m} = n$, then $\widehat{m} = n$ on $E_n$ (Lemma A.3) and the difference is 0. Otherwise $\overline{m} \le n - 1$ and, on $E_n$,

$$\overline{C}(\widehat{m}) = \overline{C}(\overline{m}) + \sum_{m=\overline{m}}^{\widehat{m}-1} \Delta\overline{C}(m).$$

Since $(\overline{\beta}_m)$ is nondecreasing, $\overline{\beta}_{m+1} - \overline{\beta}_m \ge 0$, hence

$$\Delta\overline{C}(m) = \overline{\beta}_m - (n - m - 1)(\overline{\beta}_{m+1} - \overline{\beta}_m) \le \overline{\beta}_m \le 1.$$

Thus, on $E_n$,

$$\overline{C}(\widehat{m}) \le \overline{C}(\overline{m}) + (\widehat{m} - \overline{m}) \le \overline{C}(\overline{m}) + M_n + 1,$$

where we used Step 1 to upper bound $\widehat{m}$.

**Step 4: Combine and take expectations.**   On $E_n$, by Steps 2–3 and $\overline{C}(\overline{m}) \le \overline{C}(m^\star)$,

$$\widehat{C}(\widehat{m}) \le \overline{C}(m^\star) + (M_n + 1) + b_n.$$

Also on $E_n$, $\widehat{C}(m^\star) \ge \overline{C}(m^\star) - b_n$. Deterministically, since $0 \le 1 - \beta_m \le 1$, we have $\widehat{C}(m) \le n$ for all $m$. Using $\Pr(E_n^c) \le 1/n$ gives

$$C(m^\star) = \mathbb{E}[\widehat{C}(m^\star)] \ge \overline{C}(m^\star) - b_n - 1, \qquad \mathbb{E}[\widehat{C}(\widehat{m})\mathbf{1}\{E_n^c\}] \le 1.$$

Combining,
$$C(\widehat{m}) \leq C(m^\star) + (M_n + 1) + 2b_n + 2 \leq C(m^\star) + 2R_n,$$

since $2R_n = 2M_n + 2b_n + 2n\delta_N + 2$ dominates the preceding additive terms.

In the end, we prove $R_n/C(m^\star) \to 0$.

We have $m_0 \to \infty$ and $\eta_{m_0} = 0.85\sqrt{C_n/m_0} = O(\sqrt{\log n}/n^{1/3})$, so $b_n \leq na_{m_0} = o(n)$. Finally, since $\beta^\star > 0$ is a fixed constant and $c_n = \Theta(\sqrt{\log\log n})$, we have

$$M_n = O(n^{2/3}(\log\log n)^{1/3}) = o(n).$$

Therefore $R_n = o(n)$.

Since $\overline{\beta}_m = L^{-1}(\alpha - r_m) \leq L^{-1}(\alpha) = \beta^\star$, for all $m$,

$$\overline{C}(m) = m + (n - m)(1 - \overline{\beta}_m) \geq m + (n - m)(1 - \beta^\star) \geq n(1 - \beta^\star).$$

Moreover $|C(m^\star) - \overline{C}(m^\star)| \leq b_n + n\delta_N = o(n)$ by Step 4 so $C(m^\star) = \Omega(n)$. Hence $R_n/C(m^\star) \to 0$, completing the proof. $\square$

# B. Continuous Label Spaces

For continuous label spaces, discretizing $\mathcal{Y}$ introduces an arbitrary value of $K$. When $\mathcal{Y}$ has finite measure $\nu(\mathcal{Y})$, a natural analogue of the normalized set-size term in equation 2 is the normalized measure $\nu(T_{D_m}(X))/\nu(\mathcal{Y})$. The corresponding total cost is

$$C(m) = m + (n - m)\mathbb{E}\left[\frac{\nu(T_{D_m}(X))}{\nu(\mathcal{Y})}\right].$$

The same size reparameterization used in Section 4 also applies. Let $\widetilde{Y}$ be drawn uniformly from $\mathcal{Y}$, define $S = s(X, \widetilde{Y})$, and set $\tau(\beta) = F_S^{-1}(1 - \beta)$ and

$$T_\beta(x) = \{y \in \mathcal{Y} : s(x, y) \leq \tau(\beta)\}.$$

Then

$$\mathbb{E}\left[\frac{\nu(T_\beta(X))}{\nu(\mathcal{Y})}\right] = \frac{1}{\nu(\mathcal{Y})}\mathbb{E}\left[\int_{\mathcal{Y}} \mathbf{1}\{s(X, y) \leq \tau(\beta)\}\, d\nu(y)\right] = \Pr(s(X, \widetilde{Y}) \leq \tau(\beta)) = 1 - \beta.$$

Thus the cost formulation and size reparameterization extend naturally to bounded continuous label spaces, with cardinality replaced by set measure.

# C. Proof in Section 6

**Theorem 6.2.** *Let $\hat{m}$ be the calibration set size induced by $\hat{\phi}$. When Assumption 6.1 holds, we have*

$$\mathbb{E}[Q(\hat{m})] \leq Q(m^\star) + R_n,$$

*where the regret term $R_n$ satisfies that $R_n = \tilde{O}(n^{2/3})$ and $\lim_{n\to\infty} \frac{R_n}{Q(m^\star)} = 0$.*

*Proof.* We will prove the equivalence of Assumption 6.1 and Assumption 4.4 in the PAC labeling setting. Let $B := \mathbf{1}[\widetilde{Y} = Y]$. We have

$$\frac{f_{S^*}(\tau)}{Kf_S(\tau)} = \Pr(B = 1 \mid S = \tau).$$

So Assumption 4.4 is equivalent to assume

$$\tau \mapsto \Pr(B = 1 \mid S = \tau)$$

is decreasing. And in the case of PAC labeling, if $\tilde{A} = 1$, we have $S = 0$ and if $\tilde{A} = 0$ we have $S = s(Z, 0)$. Now fix any $\tau > 0$ the event $S = \tau$ forces $\tilde{A} = 1$. Then

$$\Pr(B = 1 \mid S = \tau) = \Pr(\tilde{A} = A \mid \tilde{A} = 0, s(Z, 0) = \tau) = \Pr(A = 0 \mid s(Z, 0) = \tau).$$

Using the fact that in the PAC labeling setting, $A = 1[\hat{Y} = Y]$ and $s(Z, 0) = 1 - U$ we have it is equivalent to assume

$$u \mapsto \mathbb{E}[\mathbf{1}[Y \neq \hat{Y}] \mid U = u]$$

is increasing, which is exactly Assumption 6.1.

Applying Theorem 4.5 completes the proof. $\qquad\square$

## D. Limitations and Future Work

Our stopping rule does not explicitly use covariates to decide which examples to label beyond their role in the nonconformity score; incorporating covariate-dependent label acquisition is an interesting direction for future work.

## E. Empirical Validation of Assumptions

In this section, we provide empirical evidence supporting the monotonicity assumptions used in the theoretical analysis. All experiments and datasets follow the setup described in Section 7.

### E.1. Empirical Validation of Assumption 4.4

Assumption 4.4 requires that the density ratio $\tau \mapsto f_{S^\star}(\tau)/f_S(\tau)$ is non-increasing, where $S^\star := s(X, Y)$ denotes the nonconformity score under the true label and $S := s(X, \widetilde{Y})$ denotes the score under a uniformly sampled label $\widetilde{Y}$. Intuitively, this holds whenever the learned score function assigns low scores to true labels more reliably than to random labels.

Figure 3 empirically evaluates this assumption on ImageNet using the score $s(x, y) = 1 - \hat{p}(y \mid x)$. We estimate the densities $f_{S^\star}$ and $f_S$ using histogram-based density estimation and plot the resulting ratio as a function of $\tau$. The estimated ratio is empirically decreasing across the relevant range of scores, consistent with Assumption 4.4.

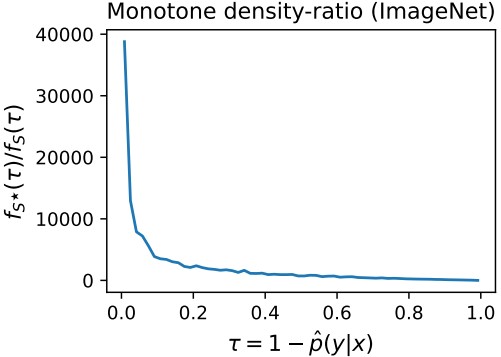

*Figure 3.* Empirical validation of the density-ratio monotonicity assumption in the conformal setting. We estimate the ratio $f_{S^\star}(\tau)/f_S(\tau)$ as a function of $\tau = 1 - \hat{p}(y \mid x)$ using histogram-based density estimates, where $S^\star$ corresponds to true-label scores and $S$ to randomly sampled labels averaged over 1000 draws. The ratio is empirically non-increasing, consistent with Assumption 4.4.

### E.2. Empirical Validation of Assumption 6.1

Assumption 6.1 requires that the conditional error rate $u \mapsto \mathbb{E}[\mathbf{1}\{Y \neq \hat{Y}\} \mid U = u]$ is increasing in uncertainty $u$. This assumption formalizes the intuition that data points where the AI model is less confident are more likely to be mislabeled.

Figure 4 evaluates this assumption across all PAC labeling datasets used in our experiments. Following the setup in Section 7, we define uncertainty as $u = 1 - \text{confidence}$, bin examples according to uncertainty, and estimate the empirical error rate within each bin. The conditional error rate empirically increases with uncertainty across datasets, consistent with Assumption 6.1.

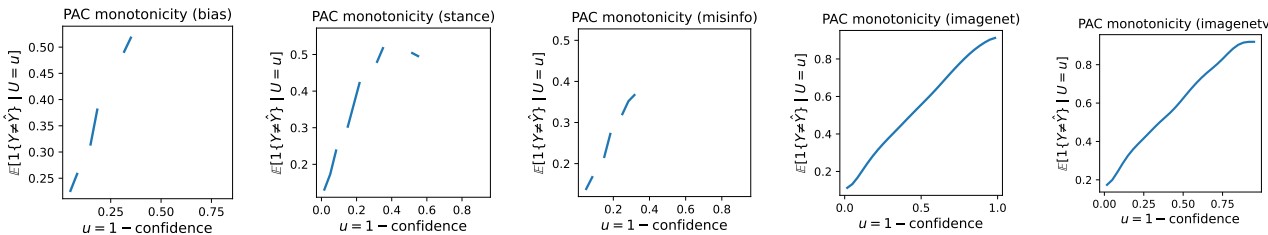

*Figure 4.* Empirical PAC monotonicity across datasets. We estimate $\mathbb{E}[1\{Y \neq \hat{Y}\} \mid U \in \text{bin}]$ as a function of uncertainty $u = 1 - \text{confidence}$ by binning $u$ and computing the empirical error within each bin, followed by Gaussian smoothing to reduce variance. The conditional error rate is increasing in uncertainty across datasets.

## F. Additional Results and Experiments

In Figure 5, we present additional plots that complement those shown in Figure 1, including results for the greedy stopping baseline described below. In particular, Figure 5(a) shows the average coverage, and Figure 5(b) shows the reduction in average prediction set size as the calibration size $m$ increases.

In Figure 6, we include complete results analogous to those in Figure 5 for the PAC labeling setting across the remaining datasets, including media bias (Baly et al., 2020), stance on global warming (Luo et al., 2020), misinformation detection (Gabriel et al., 2022), and ImageNet and ImageNet v2 (He et al., 2016). The figures also include the greedy stopping baseline described below. All datasets and experimental settings follow those used in the original PAC labeling work (Candès et al., 2025), with $\alpha = 0.1$ and $\delta = 0.05$. All results use finite-sample stabilization with $\kappa = 30$ and are averaged over 100 trials.

**Greedy stopping baseline.** As an additional practical adaptive baseline, we consider a greedy stopping rule that stops once the empirical cost no longer decreases, i.e., when $\widehat{C}(m+1) - \widehat{C}(m) \geq 0$. In practice, this rule is highly sensitive to finite-sample noise and often stops prematurely. To stabilize the rule, we instead stop only after observing $k = 5$ consecutive increases in empirical cost.

The greedy baseline is shown in red in Figures 5 and 6. While the greedy stopping rule performs reasonably well in some settings (e.g., Figures 6a, 6b, and 6c), its behavior is inconsistent across datasets. In some cases, such as Figures 6d and 6e, it can even perform worse than the fixed baseline. In other settings, it improves over the fixed baseline but still fails to match the hindsight oracle or our stopping rule (e.g., Figure 5). The greedy stopping rule is highly sensitive to noisy empirical cost estimates, which can cause it to stop either too early or too late depending on the dataset. In contrast, our stopping rule more consistently tracks the oracle calibration size and more reliably achieves lower total cost across datasets.

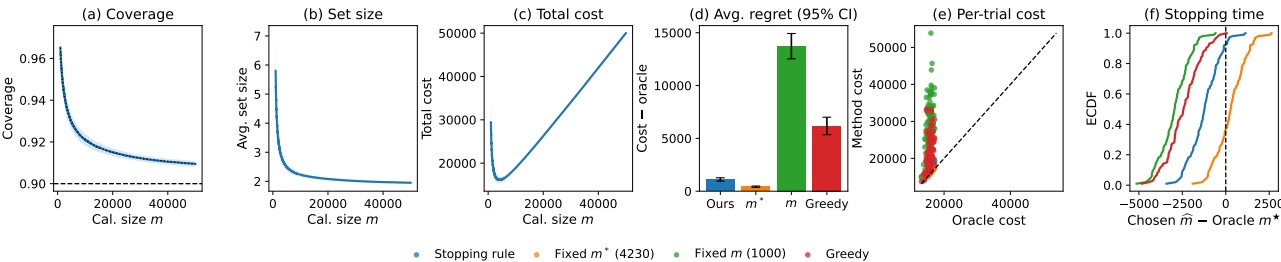

*Figure 5.* Conformal prediction full results on ImageNet. Coverage, prediction set size, total cost, regret (95% CI), per-trial cost versus the oracle, and stopping time distribution as functions of the calibration size. The stopping rule maintains valid coverage and selects calibration sizes close to the oracle optimum.

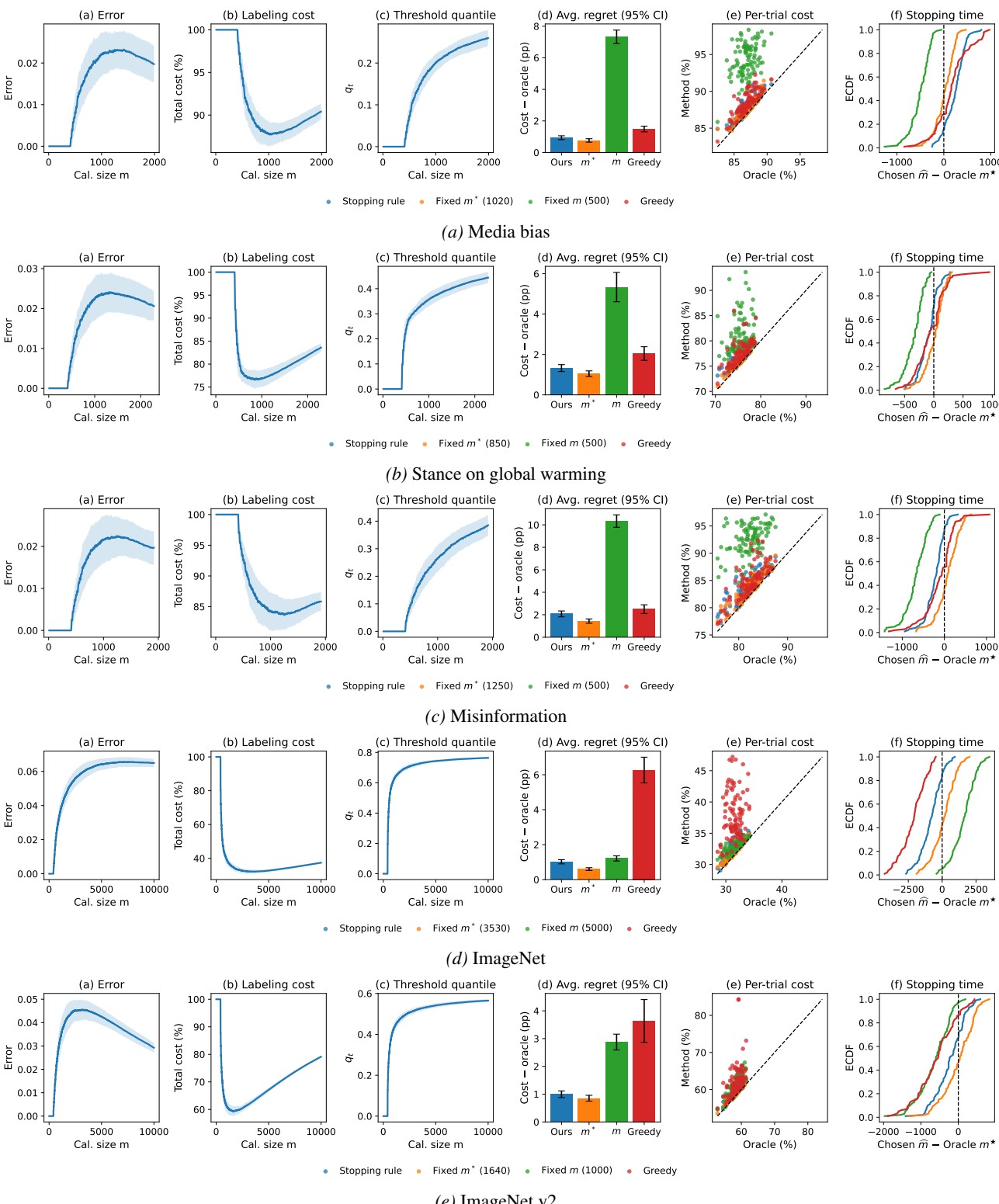

*Figure 6.* PAC labeling full results across datasets. Labeling error, total human labeling cost, threshold quantile, regret (95% CI), per-trial cost versus the oracle, and stopping time distributions for Media Bias, Stance on Global Warming, Misinformation, ImageNet, and ImageNet v2. The stopping rule consistently reduces total human labeling cost and tracks the oracle calibration size.

