# OpenReview forum: "Provably Label-Efficient Conformal Prediction"
_ICML.cc/2026/Conference — ICML 2026 regular_

### Official Review · Reviewer_Robm · 2026-03-08

**Soundness:** 3
**Presentation:** 3
**Significance:** 3
**Originality:** 3
**Overall Recommendation:** 5
**Confidence:** 3

**Summary:**

The paper considers conformal prediction with a label cost constraint. The authors set up the problem in a finite label space setting with a cost function that balances the label cost and the size of the prediction set. A O(n^{2/3}) order regret bound is established for the proposed stopping rule based on an any-time valid CP procedure and a convexity condition on the cost function. The result is also applied in a PAC labeling setting and leads to a similar regret bound in terms of labeling cost.

**Compliance With Llm Reviewing Policy:**

Affirmed.

**Final Justification:**

My questions/concerns have been addressed. I keep my score.

**Key Questions For Authors:**

1. What would be a reasonable cost function when the label space is R? One can certainly discretize a continuous label, but the discretization would lead to an arbitrary K. Some discussion along this direction would be helpful.
2. I understand that the convexity of the cost function leads to a very clean analysis. Are there interesting cases that the condition is violated? Do the results still hold under just uni-modality?
3. I would like to understand the O(n^{2/3}) rate in the regret bound. What is the heuristics of getting the O(n^{2/3}) rate? Is the rate sharp for the proposed method? Is this rate optimal among all methods?
4. I have some trouble understanding the PAC labeling framework. The second line in the input of Algorithm 3 has "uncertainties U_1, ..., u_m". Is there a typo here? What is \hat{u}_m in Eq (6)? In fact, I do not understand U_i, u_i, \hat{u}_i. Please provide more explanations and intuitions here.
5. The PAC labeling application is quite clean, which makes me wonder if the result can be applied in other distribution-free inference problems such as calibration?

**Limitations:**

Discussed above.

**Strengths And Weaknesses:**

Overall, I think this is a very clean result that has addressed a relevant problem in distribution-free inference. The stopping rule together with the regret analysis are applied to both conformal prediction and PAC labeling, which is quite neat. Though I like the paper subjectively, I am no expert in conformal prediction, and would be happy to lower my score if other reviewers have different opinions.

The paper is well written, especially the technical part in Section 5. Section 6 can still be improved with more background information (I will discuss later).

---

> ### Author Rebuttal · Authors · 2026-03-31
>
> > (Question 1) What would be a reasonable cost function when the label space is R?
>
> Thank you for this helpful question! We agree that for continuous labels, discretizing $\mathcal Y$ introduces an arbitrary $K$. For a bounded continuous label space such as $\mathcal Y=[0,1]$, a more natural efficiency term is the conformal set length $\nu(T(x))$ where $\nu(A):=\int_Ady$. The total cost can then be written as $C(m)=m+(n-m)\mathbb E[\nu(T_{D_m}(X))/\nu(\mathcal Y)]$. Moreover, the same size reparameterization used in Section 4 extends directly. Formally, recall $\widetilde{Y}$ is drawn uniformly from $\mathcal Y$, $S=s(X,\widetilde{Y})$, $\tau(\beta)=F_S^{-1}(1-\beta)$ and $T_\beta(x)=\{y\in\mathcal Y:s(x,y)\le\tau(\beta)\}$. Then
> $$\mathbb E[\nu(T_{D_m}(X))/\nu(\mathcal Y)]=\frac{1}{\nu(\mathcal Y)}\mathbb E\Bigg[\int_{\mathcal Y}1{s(X,y)\le\tau(\beta)}d\nu(y)\Bigg]=\Pr(s(X,\widetilde Y)\le\tau(\beta))=1-\beta.$$ Therefore, our analysis can be extended to such a continuous setting and we will add a discussion in the revised version.
>
> > (Question 2) I understand that the convexity of the cost function leads to a very clean analysis. Are there interesting cases that the condition is violated? Do the results still hold under just uni-modality?
>
> Thank you for the insightful question! We agree that Assumption 4.4 is mainly a sufficient condition. Our proof only uses two consequences: (i) the oracle cost is unimodal, and (ii) the miscoverage function$L^{-1}$ (Definition 4.2) is continuous on the relevant range, which is needed to control $\bar{\beta}_{m+1}-\bar{\beta}_m$ in Lemma 5.5.
>
> Therefore, the argument extends beyond Assumption 4.4 whenever these two properties hold directly. In particular, unimodality alone is not enough; continuity of $L^{-1}$ is also needed. For example, if $S \sim \mathrm{Unif}[0,1]$ and $S^* \sim \mathrm{Beta}(2,2)$, then Assumption 4.4 fails, but these two properties still hold, so our result continues to apply. We will clarify in the revision that Assumption 4.4 is a convenient sufficient condition, not the most general one.
>
> > (Question 3) What is the heuristics of getting the O(n^{2/3}) rate? Is the rate sharp for the proposed method? Is this rate optimal among all methods?
>
> We thank the reviewer for this helpful question! Since our stopping rule is based on computing a lower confidence bound on the first-order derivative of an oracle cost function, it ensures that the stopping point $\hat{m}$ exceeds the minimizer of the oracle cost function by at most $\widetilde{O}(n^{2/3})$. The regret analysis decomposes into two components: (i) an estimation error arising from replacing the oracle quantity with its empirical estimate, which decreases with $\hat{m}$, and (ii) a stopping error incurred by overshooting the oracle minimizer, which increases with $\hat{m}$. Our stopping rule balances these two terms, yielding the final regret bound of $\widetilde{O}(n^{2/3})$.
>
> As for the lower bound, we do not currently know whether this rate is sharp. A natural direction would be to seek an information-theoretic lower bound via a two-point construction: namely, to design two score distributions whose calibration streams are hard to distinguish from finitely many queried labels, yet whose oracle marginal costs differ enough to force any adaptive stopping rule to incur nontrivial excess cost. Whether this can yield a matching lower bound remains an interesting open question.
>
> > (Weakness 1 and Question 4) I have some trouble understanding the PAC labeling framework.
>
> Thank you for pointing this out! First, in Algorithm 3, $U_1,\ldots,u_m$ is a typo and should be $U_1,\ldots,U_m$. Here, $U_i$ denotes the uncertainty score for prediction $\hat{Y}_i$ on feature $X_i$, with smaller $U_i$ indicating higher confidence. Second, $\hat{u}_m$ in Eq. (6) is the learned threshold from the $m$ labeled calibration examples. It determines when to trust the AI prediction and when to query the expert. Eq. (6) chooses the largest threshold that still satisfies the target error tolerance with high probability, thereby balancing label efficiency and reliability. In the revision, we will correct the typo and add more explanation of PAC labeling framework.
>
> > (Question 5) The PAC labeling application is quite clean, which makes me wonder if the result can be applied in other distribution-free inference problems such as calibration?
>
> Thank you for this insightful comment! We do not see a direct extension to calibration, since calibration typically requires guarantees conditional on the prediction, which are stronger than the marginal guarantees of conformal prediction. That said, we view conformal risk control [1] as a promising future direction, as it generalizes conformal prediction while retaining a similar high-level perspective.
>
> [1] Angelopoulos, A.N., Bates, S., Fisch, A., Lei, L. and Schuster, T., 2022. Conformal risk control. arXiv preprint arXiv:2208.02814.

---

> > ### Author Rebuttal · Reviewer_Robm · 2026-04-03
> >
> > I thank the authors for the additional explanations.

---

### Official Review · Reviewer_hi6G · 2026-03-11

**Soundness:** 2
**Presentation:** 3
**Significance:** 3
**Originality:** 3
**Overall Recommendation:** 4
**Confidence:** 4

**Summary:**

This paper investigates cost-sensitive conformal prediction to optimize the calibration size, balancing upfront labeling expenses against downstream prediction inefficiency. The authors introduce a data-dependent stopping rule using anytime-valid confidence sequences, dynamically evaluating the marginal utility of new labels while preserving marginal coverage. Assuming a monotone density-ratio for nonconformity scores, they prove this online approach is no-regret, bounding the expected cost within $\tilde{O}(n^{2/3})$ of the optimal fixed hindsight size. Empirical evaluations on ImageNet and PAC labeling confirm that this dynamic rule closely tracks oracle performance, significantly reducing costs compared to standard static baselines.

**Compliance With Llm Reviewing Policy:**

Affirmed.

**Final Justification:**

I raise my score to 4. The rebuttal addressed my main concerns by fixing the formula inconsistency, correcting the stabilization and rerunning the experiments, adding empirical checks for the monotonicity assumptions, including a stronger adaptive baseline, and clarifying the sequential setting.

**Key Questions For Authors:**

**Questions**

See Weaknesses.

**Typo**
- line 251: "deincreasing" to "decreasing"

**Limitations:**

No. It would be helpful to discuss when the monotone density-ratio assumption may fail, the requirement of access to a large unlabeled pool and its computational burden, and possible downstream risks in high-stakes domains if the stopping rule is mis-specified or used with miscalibrated scores.

**Strengths And Weaknesses:**

**Strength.**

- The paper directly models the tradeoff between calibration labeling cost and downstream conformal efficiency through a combined objective, rather than optimizing prediction-set size for a fixed calibration budget. This makes the problem formulation practically relevant in expensive-label settings.
- The use of time-uniform concentration effectively addresses the challenge of preserving conformal validity under data-dependent stopping, and the convexity analysis under Assumption 4.4 provides a clean justification for the proposed stopping rule and the no-regret comparison to the best fixed calibration size.
- The reduction in Section 6 shows that the framework is not limited to conformal set prediction, but also yields an interpretable selective-labeling perspective with a corresponding no-regret guarantee under a transparent monotonicity condition.

**Weaknesses.**

- There are conflicting formula dependencies for the time-uniform radius $r_m$. Algorithm 2 takes $\\delta$ as a confidence budget, but Definition 4.3 and the proof of Theorem 4.1 contain expressions where $r_m$ depends on $\\alpha$ rather than $\\delta$.
- The stabilization replaces $\\alpha$ by $\\kappa / \\sqrt{n}$, which does not converge to the theoretically used $t_m=\\alpha-r_m$ for fixed $\\alpha$, contradicting the claim that the stabilization vanishes and does not affect asymptotic guarantees. Moreover, since $t_m$ appears in denominators (e.g., $2 \\eta_m / t_m$), this modification can change the stopping time, so the experiments do not directly validate the analyzed rule.
- The no-regret guarantee depends on a monotone density-ratio assumption. This is a strong structural assumption and it is not empirically checked or discussed in terms of when it holds for common conformal scores like $1-\\hat{p}(y \\mid x)$ in modern classifiers.
- In the main ImageNet experiment, the baseline is limited to a single fixed calibration size $m=1000$ plus an oracle hindsight $m^{\\ast}$. the empirical evaluation could be strengthened by including stronger practical adaptive baselines, such as (i) selecting $m$ by minimizing an estimated cost curve using a small pilot labeled set; (ii) applying a greedy heuristic that stops when the marginal empirical reduction in set size falls below the unit cost of labeling.
- The paper frames the problem as a sequential setting, but $\\beta_m$ is defined by averaging over all $n$ unlabeled points and all $K$ labels. That implies advance access to the full unlabeled pool and an $O(n K)$-style scoring burden, which is not discussed.

---

> ### Author Rebuttal · Authors · 2026-03-31
>
> > (Weakness 1) There are conflicting formula dependencies for the time-uniform radius $r_m$.
>
> Thank you for catching this. $r_m$ should be defined with $\delta$ instead of $\alpha$. We will correct it in the revision.
>
> > (Weakness 2) The stabilization replaces $\alpha$ by $\kappa/\sqrt{n}$, which does not converge to the theoretically used $t_m=\alpha-r_m$ for fixed $\alpha$, contradicting the claim that the stabilization vanishes and does not affect asymptotic guarantees.
>
> Thank you for pointing this out! We now use $t_m := \frac{\kappa}{\sqrt{n}} + \alpha - r_m$ instead of $t_m := \frac{\kappa}{\sqrt{n}} - r_m,$ and reran all experiments. This modification results in only minor changes to empirical performance and does not affect the validity of our claims. Figure 1 show the updated results under this stabilization, with the stopping rule shown in blue.
>
> > (Weakness 3) The no-regret guarantee depends on a monotone density-ratio assumption. This is a strong structural assumption and it is not empirically checked or discussed in terms of when it holds for common conformal scores.
>
> We agree that empirically checking this assumption would strengthen the paper. Figure 2 shows the empirical validation of Assumption 4.4 (monotone density ratio), namely that $\tau \mapsto \frac{f_{S^*}(\tau)}{f_S(\tau)}$ is decreasing. Figure 3 shows the empirical validation of Assumption 6.1, namely that $u \mapsto \mathbb{E}\big[ \mathbf{1}\{Y \neq \hat{Y}\} \mid U = u \big]$ is increasing across all PAC labeling datasets used in our experiments.
>
> > (Weakness 4) The empirical evaluation could be strengthened by including stronger practical adaptive baselines.
>
> Thank you for pointing this out! We now include a greedy baseline that stops when the empirical cost stops decreasing, i.e., when $\widehat{C}(m+1) - \widehat{C}(m) \ge 0$. The naive version is very sensitive to noise and stops too early, so we fix this by requiring $k$ consecutive increases in cost before stopping (we use $k=5$ in all experiments). This avoids reacting to a single noisy increase and significantly improves the performance of the greedy baseline. Results are shown in Figure 1 in red. Even with this change, the greedy baseline is inconsistent. It works reasonably well in some cases (e.g., PAC labeling on Misinfo) but poorly in others (e.g., PAC labeling on ImageNet).
>
> > (Weakness 5) The paper frames the problem as a sequential setting, but $\beta_m$
>  is defined by averaging over all $n$ unlabeled points and all $K$ labels.
>
> Thank you for highlighting this! We agree that the current exposition should more clearly distinguish sequential label acquisition from access to the unlabeled $X$. In our setting, sequentiality does not mean that future features $X$ are hidden. Rather, the sequential aspect is that labels are acquired one at a time, and each query is irreversible. Once a point is queried and labeled, it no longer requires a conformal prediction set; at the same time, the labeling cost incurred for that point cannot be recovered. This asymmetry is exactly what our objective is designed to capture: queried points incur labeling cost, while only the remaining unlabeled points contribute prediction-set inefficiency. We will revise the text to make this distinction more explicit.
>
> We also note that the burden of evaluating scores across candidate labels is not unique to our method. Multi-class conformal prediction itself constructs prediction sets by evaluating the nonconformity score over candidate labels.
>
> > (Weakness 6) Typo line 251: "deincreasing" to "decreasing".
>
> Thanks for pointing out! We will correct it in the revision.
>
>
> **Link to Figures 1-3 referenced above:** https://anonymous.4open.science/api/repo/ICML-2026-Rebuttal-BC62/file/ICML2026_Rebuttal_Figures.pdf?v=3869e089

---

> > ### Author Rebuttal · Reviewer_hi6G · 2026-04-01
> >
> > Thanks for the authors' rebuttal. I think the rebuttal has addressed all of my concerns and I raised the score to 4.

---

### Official Review · Reviewer_W9hw · 2026-03-13

**Soundness:** 3
**Presentation:** 3
**Significance:** 3
**Originality:** 3
**Overall Recommendation:** 4
**Confidence:** 4

**Summary:**

This paper studied the problem of performing split conformal prediction with an actively queried calibration set. The goal is to minimize the performance metric defined as the sum of the query cost and the size of the prediction set, which are two terms in tension. Authors proposed a stopping rule (about when to stop the query) such that when combined with (anytime-valid) split conformal prediction, it would achieve no-regret in terms of the aforementioned metric, under some regularity condition.

**Compliance With Llm Reviewing Policy:**

Affirmed.

**Final Justification:**

Most of my concerns are resolved and authors also agreed that there's some work to be done with Assumption 4.4. Hence I will keep my score.

**Key Questions For Authors:**

1. Where is $C_\delta$ used after defined in Algorithm 2?
2. What are examples where Assumption 4.4 actually holds?
3. How would the current analysis be affected if we switch to other choices of prediction sets $T_m(x)$?

**Limitations:**

It would be helpful to mention some of the limitations listed above.

**Strengths And Weaknesses:**

**Strengths:**
1. The setup of actively querying the label for the calibration set seems novel and practical. The cost chosen as a combination of number of queries and expected size of the prediction set makes sense to me as well.
2. The idea of using time-uniform confidence sequences to derive anytime/any-sample-size coverage guarantee (which implies the coverage guarantee for any stopping time) sounds novel.


**Weaknesses:**
1. Typos: "For any $m\in[n]$" in Definition 3.3 is not really needed; "deincreasing" in Assumption 4.4.
2. It seems that the form of the prediction set $T_m(x)$ is forced to be {$y: s(x,y) \leq \hat q_m$}, where $\hat q_m$ also admits the specific form as some empirical quantile of the nonconformity score over the calibration set. Since the choice of $\hat q_m$ clearly affects the performance metric $C(m)$, I would like to know if the results in this paper would also hold under other choice of $\hat q_m$ or $T_m(x)$.
3. The presentation could be probably clearer at some places. For example, in Algorithm 2 we can see that the construction of prediction sets $T_m$ and the application of stopping rule are decoupled. Theorem 4.1 is more like an anytime coverage guarantee for the sequence of sets $T_m$, which hinges on the choice of $r_m$ from the literature, while the coverage guarantee for arbitrary stopping rule is just an immediate corollary. It would be better if such points can be made clear in the context.
4. Better to have some examples where Assumption 4.4 holds.

---

> ### Author Rebuttal · Authors · 2026-03-31
>
> > (Weakness 1) Typos: "For any $m\in[n]$" in Definition 3.3 is not really needed; "deincreasing" in Assumption 4.4.
>
> Thanks for pointing out. We will correct them in the revision.
>
> > (Weakness 2 and Question 3) I would like to know if the results in this paper would also hold under other choice of $\hat{q}_m$ or $T_m(x)$.
>
> Thank you for this helpful question! We agree that changing $\hat{q}_m$ can change the induced cost curve $C(m)$. However, we believe that the analysis is not tied to split conformal prediction per se. We adopt split conformal prediction because it is the standard, widely used conformal framework [1], and it provides a clean way to formulate the stopping problem.
>
> More importantly, our analysis relies only on two structural properties: (i) the prediction set is generated by thresholding a scalar score, and (ii) the threshold is computed from an i.i.d. calibration set. These properties are not unique to split conformal. In particular, APS [2] and RAPS [3] also construct prediction sets by applying a scalar threshold selected from an i.i.d. calibration set, so the same style of analysis should extend to those methods as well.
>
> We agree that moving beyond threshold-based choices is an interesting and nontrivial direction for future work, and we will clarify this point in the revision.
>
> > (Weakness 3) The presentation could be probably clearer at some places.
>
> We agree, and this is a very helpful suggestion. Theorem 4.1 first establishes an anytime-valid coverage guarantee for the full sequence $(T_m)_{m=1}^n$ under an arbitrary stopping rule. The coverage guarantee at the stopped time $\hat{m}$ then follows immediately as a corollary of this stronger statement. We will revise the text to make this two-step structure more explicit.
>
> > (Weakness 4 and Question 2) What are examples where Assumption 4.4 actually holds?
>
> Thank you for bringing this up! Whether this assumption holds depends on the quality of the nonconformity score function, for which we provide empirical examples. Specifically, Figure 2 empirically examines Assumption 4.4, showing that the map
> $
> \tau \mapsto \frac{f_{S^*}(\tau)}{f_S(\tau)}
> $
> is decreasing in our experiments.
>
> Link to Figure 2 referenced above: https://anonymous.4open.science/api/repo/ICML-2026-Rebuttal-BC62/file/ICML2026_Rebuttal_Figures.pdf?v=3869e089
>
> > (Question 1) Where is $C_\delta$ used after defined in Algorithm 2?
>
> Thank you for catching this. As written, $C_\delta$ is introduced in Algorithm 2 but is not used afterward in the procedure: the algorithm proceeds using only $u_m = (1-\alpha) + r_m$, where $r_m$ is defined directly in the text below the algorithm. This is leftover notation from the derivation of the time-uniform confidence bonus. We will correct it in the revision.
>
> [1] Angelopoulos, A.N. and Bates, S., 2023. Conformal prediction: A gentle introduction. Foundations and Trends in Machine Learning, 16(4), pp.494-591.
>
> [2] Romano, Y., Sesia, M. and Candes, E., 2020. Classification with valid and adaptive coverage. Advances in neural information processing systems, 33, pp.3581-3591.
>
> [3] Angelopoulos, A., Bates, S., Malik, J. and Jordan, M.I., 2020. Uncertainty sets for image classifiers using conformal prediction. arXiv preprint arXiv:2009.14193.

---

> > ### Author Rebuttal · Reviewer_W9hw · 2026-04-03
> >
> > I thank authors for their detailed explanations. I think my questions are mostly resolved while there's still some minor concern around the relevance of Assumption 4.4 since it seems never appeared in the literature before (as there's no citation). I would like to keep my score at this point.

---

> > > ### Author Response · Authors · 2026-04-05
> > >
> > > We thank the reviewer for this thoughtful feedback. Regarding the question about Assumption 4.4, we agree that this assumption is novel in the literature. We view our work as a first step in introducing and studying this assumption, and our empirical results suggest that, for the datasets we consider, the corresponding map is indeed decreasing in most experiments. Moreover, the algorithm derived under this assumption consistently achieves a lower cost than the baselines in our experiments.
> > >
> > > We also appreciate the reviewer for raising this point again and agree that an interesting direction for future work is to identify alternative assumptions under which cost minimization remains feasible. We will include this discussion in the final version.

---

### Official Review · Reviewer_ooog · 2026-03-18

**Soundness:** 4
**Presentation:** 4
**Significance:** 3
**Originality:** 4
**Overall Recommendation:** 5
**Confidence:** 4

**Summary:**

This work studies the problem of average set size efficiency of conformal methods, with the additional focus on the cost of querying gold standard labels for calibration set. They formalize the problem by defining a cost function that penalize for both the final set size of the sets, and the number of queried data points. Consequently, the formalize an algorithmic framework, in which different stopping rules can be evaluated via a notion of regret. Then they provide a finite sample algorithm that provable satisfies a distribution free coverage guarantee and an asymptotic no-regret criteria. They also showcase the validity of their theoretical guarantees through a number of numerical results.

**Compliance With Llm Reviewing Policy:**

Affirmed.

**Key Questions For Authors:**

.

**Limitations:**

yes

**Strengths And Weaknesses:**

- The problem of set size efficiency is of the special interest of CP community. This paper brings a new flavor to this research direction that, to the best of my knowledge, is novel. I find their mathematical formulation of the problem clean and elegant. Their algorithm and their theoretical analysis is also sound and intuitive.

 - The authors did not answer/focus on a number of natural questions that I find important for a better understanding of the structure of the problem they proposed. Looking at equation 2, where the cost function is defined, a natural question is what is the minimizer m^*, if solved over population. In other words, how complex m^* is as a function of the joint distribution of X and Y? Is it possible to write down a closed form for it? To answer to that might open up room for alternative algorithmic solutions for the problem, and also, a deeper understanding of how hard the problem actually is. On the same note, in Section 4.2, the authors introduce a specific parametrization for the prediction sets. The same question applies there too. What is the optimal parametrization of the prediction sets, if the goal is to minimize the cost function provided in equation 2? Is the specific parametrization used in Section 4.2 aiming directly to minimize the cost?

- It appears that the algorithm is treating the data points as if like they are all the same. In other words, the algorithm is not using the unique covariates of each data point. I understand that 1) to guarantee marginal coverage covariates are not needed and 2) using covariates can actually break the symmetry of the algorithm and make the analysis harder. However, the covariates should be useful in this scenario. Intuitively, asking the labels where uncertainty is higher, perhaps is more valuable than in other places, and this should be tracked with the help of covariates. Is there anyway to incorporate this in your algorithmic framework? At the very least, this is worth discussing as future work.

---

> ### Author Rebuttal · Authors · 2026-03-31
>
> > (Weakness 1) Looking at equation 2, where the cost function is defined, a natural question is what is the minimizer m^, if solved over population. In other words, how complex m^ is as a function of the joint distribution of X and Y? Is it possible to write down a closed form for it?
>
> Thank you for raising this structural question! We agree that the population minimizer of Equation 2 deserves further clarification. We believe no closed-form solution exists for the population minimizer. To see why, recall $S^\star=s(X,Y)$ and $S=s(X,\widetilde{Y})$ where $\widetilde{Y}\sim\mathrm{Unif}(\mathcal Y)$. Let $F_S$ and $F_{S^\star}$ denote the CDF of $S$ and $S^\star$. Algorithm 1 selects $\hat{q}_m$ as the empirical $(1-\alpha+r_m)$-quantile of the nonconformity scores $s(X_i,Y_i)$; equivalently, $\hat{q}_m$ is the $k_m=\lceil m(1-\alpha+r_m)\rceil$-th order statistic of $S^\star_1,\cdots,S^\star_m$. Substituting into Equation 2, the cost function can be rewritten as
>
> $$
> C(m)=m+(n-m){\mathbb E}\Big[F_S\Big(S^\star_{(k_m)}\Big)\Big].
> $$
> This expression involves the expectation of $F_S$ evaluated at the $k_m$-th order statistic of $m$ i.i.d. draws from $F_{S^\star}$, where the index $k_m$ itself depends on $m$ through a ceiling function. The resulting dependence on $m$ is highly nonlinear and intertwines properties of both $F_S$ and $F_{S^\star}$, precluding a universal closed-form minimizer. Thus, without additional parametric assumptions on the distribution $\mathcal D$ of $(X,Y)$ and on the structure of the nonconformity function $s$, the population minimizer does not have an analytic formula.
>
> We further emphasize that our framework does not assume access to the underlying distribution $\mathcal D$; the analysis operates entirely with i.i.d. samples. This is precisely what motivates the adaptive, sample-based stopping rule proposed in Section 4.
>
> > (Weakness 2) What is the optimal parametrization of the prediction sets, if the goal is to minimize the cost function provided in equation 2? Is the specific parametrization used in Section 4.2 aiming directly to minimize the cost?
>
> Thank you for this thoughtful question! Algorithm 2 uses the split conformal prediction set family $T_q(x)=\{y:s(x,y)\le q\}$. Section 4.2 does not introduce a new family; it simply re-parameterizes the same sets via $q=\tau(\beta)=F_S^{-1}(1-\beta)$. Thus, optimizing over $q$ or $\beta$ gives the same predictor. We use the $\beta$ parameterization because it makes the normalized expected set size linear, which simplifies the analysis.
>
> Meanwhile, we would like to note that our goal is to choose the calibration size subject to a high-probability coverage constraint, not to perform unconstrained minimization of Eq. (2) over all measurable set-valued predictors. Accordingly, our guarantees are twofold: validity (Theorem 4.1) and no regret (Theorem 4.5). We view this as complementary to prior work that optimizes set construction for a fixed calibration size.
>
> >  (Weakness 3) Intuitively, asking the labels where uncertainty is higher, perhaps is more valuable than in other places, and this should be tracked with the help of covariates. Is there anyway to incorporate this in your algorithmic framework?
>
> Thank you for the question! One potential challenge is that a covariate-dependent stopping rule or conformal predictor would itself need to be learned, unless one adopts a fixed heuristic. In that case, the sample complexity would depend on the complexity of the underlying hypothesis class, which may be substantially larger than the anytime-valid correction used in our current split conformal procedure. As a result, additional labeled data would generally be needed to optimize over such a class reliably. Work on length optimization in conformal prediction [1] is related in spirit to this type of extension.
>
> At the same time, we would like to note that our framework is already quite general: covariate information can be incorporated directly into the design of the nonconformity score. Once the score is fixed, the stopping rule can still treat all data points symmetrically, while the effect of the covariates is reflected through the score itself. In this sense, the use of covariates can be handled at the score-design stage without requiring a separate covariate-dependent stopping rule.
>
> We thank the reviewer again for raising this point. We agree that this is an interesting direction for future work, and we will add a discussion in the final version.
>
> [1] Kiyani, S., Pappas, G. and Hassani, H., 2024. Length optimization in conformal prediction. Advances in Neural Information Processing Systems, 37, pp.99519-99563.

---

> > ### Author Rebuttal · Reviewer_ooog · 2026-04-07
> >
> > I keep my positive score. Thanks for the answers.

---

### Decision · Program_Chairs · 2026-04-30

**Decision:**

Accept (regular)

**Comment:**

The paper presents methods for label-efficient calibration in conformation prediction. In particular, the techniques presented account for the trade-off between the cost related to labeling calibration samples and the size of the subsequent prediction sets.

All the reviewers appreciated the well-funded and theoretically backed new methods proposed. In addition, the addressed topic of conformal prediction with limited labeling information is very relevant, so that the methods introduced can be of interest for the research community.

One possible limitation of the paper is that the number of labeled calibration samples may have a small effect on the size of prediction sets. For instance, other factors such as the nonconformal score used may have a more direct effect on the size of prediction sets. Having said that, the positive aspects of the paper outweigh its limitations, and the methods presented represent a meaningful contribution to the field.